# Unveiling Environmental Sensitivity of Individual Gains in Influence Maximization

Xinyan Su[1,2], Zhiheng Zhang[3,4]*, Jiyan Qiu[1,2], Zhaojuan Yue[1], Jun Li[1]*

[1]Computer Network Information Center, Chinese Academy of Sciences, Beijing, China
[2]University of Chinese Academy of Sciences, Beijing, China
[3]School of Statistics and Data Science, Shanghai University of Finance and Economics,
Shanghai 200433, P.R. China
[4]Institute of Data Science and Statistics, Shanghai University of Finance and Economics,
Shanghai 200433, P.R. China
{suxinyan,qiujiyan,yuezhaojuan,lijun}@cnic.cn
zhangzhiheng@mail.shufe.edu.cn

## Abstract

Influence Maximization (IM) seeks a seed set to maximize information dissemination in a network. Elegant IM algorithms could naturally extend to cases where each node is equipped with a specific weight, reflecting individual gains to measure its importance. In prevailing literature, these gains are typically assumed to remain constant throughout diffusion and are solvable through explicit formulas based on node characteristics and network topology. However, this assumption is not always feasible due to two key challenges: 1) *Unobservability*: The individual gains of each node are primarily evaluated by the difference between the outputs in the activated and non-activated states. In practice, we can only observe one of these states, with the other remaining unobservable post-propagation. 2) *Environmental sensitivity*: Beyond nodes' inherent properties, individual gains are also sensitive to the activation status of surrounding nodes, which change dynamically during propagation even when the network topology is fixed. To address these uncertainties, we introduce a Causal Influence Maximization (CauIM) framework, leveraging causal inference techniques to model dynamic individual gains. We propose two algorithms, G-CauIM and A-CauIM, where the latter incorporates a novel acceleration technique. Theoretically, we establish the generalized lower bound of influence spread and provide robustness analysis. Empirically, experiments on synthetic and real-world datasets validate the effectiveness and reliability of our approach.

## 1 Introduction

Information propagation over networks has been booming in recent years. Due to the power of the "word-of-mouth" phenomenon, influence spread has proven to be essential in various applications, such as viral marketing [7], HIV prevention [55], and recommendations [10]. The problem of selecting the seed set to maximize information spread is known as the **I**nfluence **M**aximization (IM) [28].

Beyond optimizing the total number of infected nodes, current research has focused on investigating the individual gains of each node in real-world scenarios, referred to as weighted-IM [52, 53, 19]. Researchers endeavor to address the question: *how can limited resources be utilized to maximize total gains?* This challenge manifests in various network scenarios, such as student networks and email networks, involving activities like awareness dissemination and product promotion. For example,

---

*Corresponding Authors.

39th Conference on Neural Information Processing Systems (NeurIPS 2025).

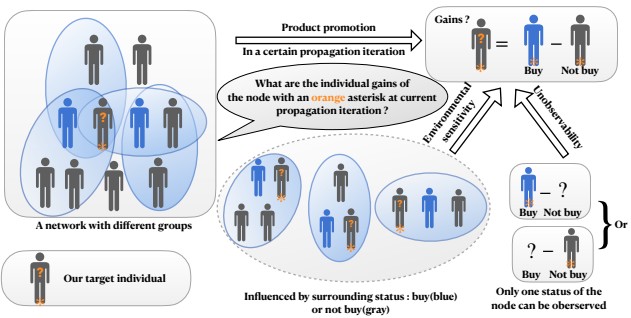

Figure 1: Illustration of individual gains during a certain propagation iteration in a product promotion scenario, focusing on the starred (*) node. The leftmost part represents an iteration state of the network represented as hyperedges (blue ovals). Each node is either activated (blue), indicating a purchase behavior, or inactive (gray). The orange star indicates the target individual whose gain we aim to evaluate, and "?" represents an unknown status. The middle panel illustrates the key question at the current propagation step: *"if we promote now, what gain would the starred node contribute?"* The individual gain is defined as the counterfactual difference $Y_i(\text{buy}) - Y_i(\text{not buy})$, representing the difference between the profit of the node in its activated and its non-activated state. Two core challenges are highlighted: *unobservability*, i.e., only one status of each node can be observed, with the counterfactual scenario unknown, and *environmental sensitivity*, which indicates the individual gains of a node are influenced by the activation status of surrounding nodes.

when targeting users with varying purchasing power in product promotion, these users exhibit diverse purchasing behaviors, resulting in varying profits for the seller. Here, by regarding purchasing power as individual gains, the goal is to identify specific users for product advertising and optimize the overall difference in profit gains pre- and post-product promotion dissemination.

Researchers usually assume that the purchasing power of each node remains observable and stable [29] throughout the entire process. Such a weighted IM setting appears to be a natural extension of traditional IM and hence leads to relatively limited exploration [28]. However, in practice, this setting would be violated, and we summarize it as two fundamental properties, as illustrated in Fig. 1 : *(i) Unobservability*. Accurately quantifying the actual purchasing power of each user is hindered by the limited observations of purchase occurrences (where only two outcomes are observable: activated or not, corresponding to purchase or non-purchase for each node; one represents the "factual", and the other "counterfactual"), thereby complicating the determination of the actual increase in benefit gains that each user can deliver to sellers; *(ii) Environmental sensitivity*. The expected purchasing power of each person is not only associated with the individuals themselves but also influenced by the attitudes of their social contacts. For example, as more people in a friend circle make purchases, individuals become increasingly susceptible to influence and are likely to make additional purchases. These properties indicate that individual gains in IM are environmentally dynamic and challenging to ascertain, posing challenges in their computation.

To tackle these challenges, we employ causality techniques to model the advertising problem, inspired by the concept of Individual Treatment Effect (ITE) [45]. It measures the disparity between the factual world and the counterfactual one, which goes beyond traditional observational studies, such as those based on network structure or direct assignment of other feature weights [28, 14]. By incorporating the process of inference, we transition the problem from observational study to the direct utilization of ground truth for measuring individual gains at the node level. Drawing from this, we propose the **Cau**sal **I**nfluence **M**aximization (CauIM) framework. Specifically, in the hypergraph modeling[2], we redefine the objective function of traditional IM by incorporating ITE as weights assigned to each node. This formulation integrates both internal covariate information and external environmental information for each node (Fig. 1).

---

[2]The utilization of IM in hypergraphs [2, 56] introduces a higher-order structure that establishes connections among clusters, effectively reflecting real-life relationships in general graphs, and treating the traditional normal graph as a special case.

Noteworthy, it is not merely a simple causality-plug-in interdisciplinary attempt since we should proactively challenge the Stable Unit Treatment Value Assumption (SUTVA) in causality [1], which permits the presence of interference between different nodes as illustrated above. Taking a step forward, even if there is pioneering exploration upon interference-based causality [37, 36, 32], distinguishing CauIM from such literature, we delve deeper into practical constraints: the original treatment policy can solely influence a "limited seed set", and then the causality estimator should be considered under a propagation process that involves temporal instability and computational burden. This process is highly non-trivial due to the dynamic environmental information associated with each node. In this context, we further provide an efficient and stable computational enhancement, which replaces the conventional greedy selection with vectorized derivative-based operations to improve scalability in practice.

In sum, our contributions are summarized as follows: **i)** We propose CauIM, a novel framework for influence maximization in networks that incorporates community structures and environment-sensitive individual node effects, providing a new perspective on modeling dynamic individual gains in networks. We also justify its practical applicability in real-world scenarios.**ii)** We demonstrate the effectiveness (approximate optimal guarantee), robustness, and acceleration feasibility of CauIM, particularly when ITE values are not strictly positive or suffer from estimation bias. In correspondence, we provide the greedy-based implementation (G-CauIM) and further design the Gradient-based Accelerated CauIM (A-CauIM). **iii)** We conduct experiments on three real-world datasets and one synthetic dataset. It not only supports our theoretical claim upon effectiveness and robustness but also validates the efficiency improvement of A-CauIM in practice.

## 2 Related work

**Influence Maximization (IM)** IM is first identified as an algorithm problem by Kempe in [28] and has given rise to several notable variants, including simulation-based (CELF [31], CELF++ [16]), sketch-based (RIS [5], TIM [48], IMM [49]), and heuristic algorithms (HADP [56]), etc. Three key elements of the problem are 1) graph structure, 2) diffusion process and 3) seed selection. For instance, CELF, RIS, and TIM emphasize iterative algorithmic enhancements while ensuring theoretical assurances within triggering models. In contrast, the latest heuristic algorithm, HADP, prioritizes computational efficiency at the cost of sacrificing theoretical guarantees. Additionally, recent learning-based IM methods [6, 30, 35] focus on understanding the inherent nature of individual node representations concerning the marginal influence gain. Nevertheless, these methods exhibit limitations in model generalization and the reliability of final results. The original optimization objective requires reassessment in diverse scenarios. Dynamic IM studies, such as Peng [44], focus on evolving edge probabilities while maintaining submodularity guarantees. Their objective differs from ours: the edge set changes over time, whereas we consider static networks where node rewards vary with unobserved environmental factors and treatment assignments. Hence, these methods are orthogonal to our framework. Researchers have increasingly focused on exploring the heterogeneity of importance between nodes based on these methods [52, 23, 15]. Nevertheless, most existing studies rely on fixed topological or attribute-based priors and treat individual importance as constant, lacking a unified framework that accounts for environmental sensitivity when estimating dynamic individual effects. We refer readers to the Appendix I for more details.

**Treatment effect estimation** How to recover the ITE directly from the observational data instead of randomization test [46] is currently receiving a lot of attention. There are two main strategies for estimation: 1) weighting-based methods [33, 34, 12], and 2) representation-based methods [47, 26]. In this paper, we follow the second strategy. Ma et al. [37] estimated the causal effect via representation learning on the more general hypergraph. However, these methods do not consider the IM question.

## 3 Preliminary

**Notations and Basic Concepts** We develop our model on an undirected hypergraph $\mathcal{G}(\mathscr{V}, \mathscr{H}, \mathbb{H})$, where $\mathscr{V} := \{v_1, v_2, ...v_n\}$, $\mathscr{H} := \{h_1, h_2, ...h_m\}$, representing the node set and the hyperedge set respectively, and $\mathbb{H} \in \{0, 1\}^{m \times n}$ denotes the incidence matrix between hyperedges and nodes. Each undirected hyperedge represents a social connection among the nodes it contains. For each

node $v_i$, covariate $X_{v_i}$ denotes its node feature and $\mathcal{N}_{v_i}$ indicates the set of its neighborhood[3]. To maintain clarity, we abuse the notation that $X_{v_i} = X_i$ and $N_{v_i} = N_i$, applying the same convention for symbols with the subscript $v_i$.

Stepping forward, we would like to introduce several broad concepts in causal inference: the potential outcome (individual profit under one status) of each node $v_i$ is denoted as $Y_i(T_i = t; X_i, \boldsymbol{T}_{-i}, \boldsymbol{X}_{-i})$. Here $t = 0, 1$ refers to the case where node $v_i$ is activated in the diffusion process or not[4] Moreover, $\boldsymbol{T}_{-i}, \boldsymbol{X}_{-i}$ represent the environmental information, namely,

$$
\begin{aligned}
\boldsymbol{T}_{-i} &:= \{T_1, T_2, ...T_{i-1}, T_{i+1}, ...T_n\}, \\
\boldsymbol{X}_{-i} &:= \{X_1, X_2, ...X_{i-1}, X_{i+1}, ...X_n\}.
\end{aligned}
\tag{1}
$$

The individual treatment effect (ITE) is defined as

$$
\begin{aligned}
\tau_i &:= Y_i(T_i = 1; X_i, \boldsymbol{T}_{-i}, \boldsymbol{X}_{-i}) \\
&\quad - Y_i(T_i = 0; X_i, \boldsymbol{T}_{-i}, \boldsymbol{X}_{-i}).
\end{aligned}
\tag{2}
$$

Here "treatment" $\{T_i, \boldsymbol{T}_{-i}\}$ indicate the activation status of $v_i$ and its surrounding nodes. ITE represents the difference in node outcomes between the activated and inactivated cases. As illustrated in the introduction, it cannot be directly extracted from observations (*property (i)*), and it also depends on the activation state of surrounding nodes (*property (ii)*).

**Problem Formulation**  We adopt the widely-used Susceptible-Infected Contact Process (SICP) diffusion model [56]. Starting from an initial seed set $S_0$, diffusion unfolds in discrete steps. In each iteration, every activated node $v$ randomly selects one of its affiliated hyperedges and attempts to activate its inactive neighbors $u \in \mathcal{N}_v$ within that hyperedge, each with a certain probability. This process continues until no new activations occur, as illustrated in Appendix Fig.4. Our framework also supports alternative diffusion models, such as Linear Threshold (LT) model [18, 17], which can be seamlessly integrated with the algorithms presented in Section 4.

We define $ap(v_i; S_0)$ as the probability that node $v_i$ gets infected in the entire propagation process initiated by seed set $S_0$. According to Wang et al. [52], $ap(v_i; S_0) = \sum_{u \in S_0} p_r(u, v_i)$, where $p_r(u, v_i)$ denotes the probability of reachability from node $u$ to $v_i$ inclusive of all reachable paths. Finally, we identify the objective function as the expected total causal influence during the diffusion:

$$
\sigma(S) = \mathbb{E}\left[\sum_{v_i \in \mathcal{V}} ap(v_i; S)\tau_i\right], S \subseteq \mathcal{V}.
\tag{3}
$$

Here $\tau_i$ is identified in Eq. 2 and the expectation takes upon activation status $\{T_i, \boldsymbol{T}_{-i}\}$ in all possible propagation process. The goal of CauIM is to find

$$
S^* = \text{argmax}_S\{\sigma(S)\}, s.t. |S| \leq K.
\tag{4}
$$

where $K$ is a fixed budget. Compared with the traditional definition, this formulation provides a concise yet general expression at the cost of introducing the relatively difficult-to-calculate term $ap(\cdot)$. Due to its computational difficulty, we will elaborate on an efficient approximation and acceleration process in the next section.

**Assumption 3.1** (Bounded ITE and Consistency). 1) Bounded ITE: $\max_{v_i \in \mathcal{V}} |\tau_i| \leq M$, where $M$ is a constant. 2) Consistency [9]: Potential outcomes $Y_i(T_i = t; X_i, \boldsymbol{T}_{-i}, \boldsymbol{X}_{-i})$ are deterministic in Eq. 2 and equal to the observational values of $Y$ for $t = 0, 1$ given fixed $\{X_i, \boldsymbol{T}_{-i}, \boldsymbol{X}_{-i}\}$.

These are standard assumptions for ITE in causal inference [43]. Notably, Assumption 3.1 accounts for interference effects that violate SUTVA [24], i.e., "the potential outcomes for any unit do not vary with the treatment assigned to other units". We incorporate such effects into an *environment function*: $O_i = \text{ENV}(\mathbb{H}, \boldsymbol{T}_{-i}, \boldsymbol{X}_{-i})$. This function follows [37], where environmental covariates and treatment assignment are summarized as a low-dimensional representation. We formalize this concept in Assumption 3.2.

---

[3]Here $v_i$ and $v_j$ are neighborhoods, indicating that they co-occur in at least one hyperedge.

[4]Here $Y(\cdot)$ is a function of $\{T_i; X_i, \boldsymbol{T}_{-i}, \boldsymbol{X}_{-i}\}$. The first two items refer to the inherent information of node $v_i$, and the last two items refer to the environmental information. We omit the information $\mathbb{H}$ in the mapping process since it remains stable in this paper. We defer the extension to dynamic graphs to future research.

**Assumption 3.2** (Environment Assumption under Interference [37])**.** For each node $v_i$, the two potential outcomes in Eq. 2 are conditionally independent given $\{T_i, O_i\}$.

Assumption 3.2 extends the standard ignorability assumption [25] to the graphical case, where the authors claimed that the pair of potential outcomes is independent of the treatment assignment, given the covariates of each node. It guarantees there are no unmeasured covariates in the graph, which is a fairly broad assumption and has been adopted by Ma et al. [37]. Assumption 3.2 essentially ensures that the two effects ($\{X_i, T_i\}, \{\boldsymbol{X}_{-i}, \boldsymbol{T}_{-i}\}$) together constitute a sufficient statistic for ITE. In other words, ITE could be legitimately estimated via observations under these assumptions and hence provides the potential to design the estimator [5].

## 4 Methodology

In this section, we first identify three primary challenges when designing our algorithms. Subsequently, we provide a detailed introduction to two algorithms within our CauIM framework. The first is an offline greedy-based implementation for causal influence maximization named G-CauIM (Alg. 1). We then improve the efficiency of G-CauIM by speeding up the diffusion and greedy selection process, and propose a Gradient-based Accelerated CauIM, referred to as A-CauIM (Fig. 2). Finally, we theoretically demonstrate the algorithm's effectiveness. Key notations for this section can be found in Appendix 3. Notably, classic IM maximizes a monotone submodular count of activations under an implicit assumption of fixed, context-independent, nonnegative marginal gains. In CauIM, node-level gains are weighted and context dependent (they may vary with neighbors' states and be negative), so submodularity may break, and sketch-based pipelines [49, 48] offer no guarantees. We thus begin by summarizing three main challenges as follows.

---

**Algorithm 1:** G-CauIM

**Input:** $\mathcal{G}(\mathscr{V}, \mathscr{H}, \mathbb{H})$; seed number $K$; $X_i$, initial treatment $T_i$ and $\boldsymbol{T}_{-i}$ of each node $v_i$; observational data
$\quad D = \{Y_i(t; \cdot)\}_{v_i \in \mathscr{V}}$, where $t = T_i$; the ITE bound $M$.

**Output:** Deterministic seed set $S^*$ with $|S^*| = K$.

1   **Function** $\widehat{\text{ITE}}$*($X_i$, $T_i$, $\boldsymbol{T}_{-i}$, $\mathcal{G}$; $\theta$)***:**

2     Compute the representation $Z_i$ of $X_i$ via representation learning;

3     Compute the high-order interference representation $O_i := \text{ENV}(\mathbb{H}, \boldsymbol{T}_{-i}, \boldsymbol{Z}_{-i}; \theta)$ (Assumption 3.1 and Assumption 3.2 );

4     Concatenate $Z_i$, $O_i$ and feed them into a Multi-Layer Perceptron (MLP):
$\quad \{\hat{Y}_i(1; \cdot), \hat{Y}_i(0; \cdot)\} \sim \text{MLP}([Z_i || O_i]);$

5     Compute the ITE $\hat{\tau}_i = \hat{Y}_i(1; \cdot) - \hat{Y}_i(0; \cdot)$ for $v_i$;

6     **return** $\hat{\tau}_i$*;*

7   **Function** *Main***:**

8     (Initialization) $S^* = \emptyset$; Loss $= 0$;

9     (Training) Using the above $\widehat{\text{ITE}}(\cdot; \theta)$ function, compute the cumulative loss by $D$:
$\quad \text{Loss} = \sum_{v_i \in \mathscr{V}, t=0,1} \left| (\hat{Y}_i(t; \cdot) - Y_i(t; \cdot)) \mathbb{I}(T_i = t) \right|$ (only the factual term is active via $\mathbb{I}(T_i = t)$) ;

10     (Projection to bounded-ITE set) Define the feasible set $\Theta_M := \{\theta : \max_{v_i \in \mathscr{V}} |\hat{\tau}_i(\theta)| \leq M\}$. Set
$\quad \theta' := \Pi_{\Theta_M}(\theta^{opt}) := \arg\min_{\theta \in \Theta_M} \|\theta - \theta^{opt}\|_2.$

11     **for** $|S^*| < K$ **do**

12       Conduct propagation under current seed set $S^*$, generate $\hat{\tau}_i = \widehat{\text{ITE}}(X_i, T_i, \boldsymbol{T}_{-i}, \mathcal{G}; \theta')$ for $v_i \notin S^*$, where $T_i$ is changed to its current activated state( 0 or 1), and $\boldsymbol{T}_{-i}$ is changed based on other nodes' activated states, $\theta' := \theta^{opt} + \triangle_\theta$, $\triangle_\theta := \min\{\|\theta_q\| : \exists \delta \leq \|\theta_q\|, \widehat{\text{ITE}}(\cdot; \theta + \delta) \leq M\}$, repeat the process and get the mean;

13       $v_0 = \arg\max_{v \notin S^*} \{\sigma(S^* \cup \{v\}) - \sigma(S^*)\}$;

14       $S^* = S^* \cup \{v_0\}$;

15   **return** $S^*$.

---

[5] Under these two assumptions, it has been demonstrated that the expected potential outcome of $v_i$ could be computed by observational data [37], namely, $\mathbb{E}(Y_i(T_i = t; X_i, \boldsymbol{T}_{-i}, \boldsymbol{X}_{-i})) = \mathbb{E}(Y_i \mid X_i = x_i, T_i = t, o_i), t \in \{0, 1\}$.

## 4.1 Three Challenges

(i) Unmeasured individual effect(ITE) (Eq. 2): the inherent limitations in causal inference necessitate the recovery of the counterfactual to address the "missing data problem" in our objective function Eq. 4. Furthermore, it may vary across iterations due to its dependence on environmental information $\boldsymbol{X}_{-i}, \boldsymbol{T}_{-i}$. (ii) Approximate optimal guarantee: The traditional greedy-based IM might not guarantee sub-optimal properties due to the unknown individual effect as mentioned above and, therefore, requires re-analysis. (iii) Estimation bias: CauIM exhibits robustness against biases stemming from individual effect estimation and the sampling strategy.

*Address challenge (i)*: ITE estimation. Motivated by Ma et al. [37], Ma and Tresp [38], we recover the individual ITE from observational data using a neural network model represented by $\widehat{\mathrm{ITE}}(\cdot)$ function in both G-CauIM and A-CauIM. To handle dynamic characteristics of ITE, we incorporate a model parameter adjustment strategy into Alg. 1 of G-CauIM. Additionally, we employ an approximation strategy in A-CauIM, as depicted in the ITE Estimator section of Fig. 2. Further details of this procedure are provided in the subsequent Algorithms section.

*Address challenge (ii)*: To achieve approximate optimum, we inductively select the seed candidate via greedy search. We show this greedy strategy will also hold a weaker but analogous order of $(1 - \frac{1}{e})$ approximate optimum level of the traditional IM in further theoretical parts:

$$v_0 = argmax_{v \notin S^*} \left\{ \sigma \left( S^* \cup \{v\} \right) - \sigma(S^*) \right\}. \tag{5}$$

It can be seen as a more general result since the traditional IM serves as the special case with $\tau = 1$. However, the analysis of the above settings is harder since the submodularity might not always exist due to the potential negative ITE. Additionally, for practical issues, We refer readers to I for the Monte Carlo-based greedy CauIM.

*Address challenge (iii)*: Our CauIM model demonstrates robustness against bias in estimating Individual Treatment Effects (ITE), which will be outlined in the theoretical discussion. This resilience is ensured under broad assumptions regarding the controlled probability $p_r(v, u)$ of reaching all nodes $u \in \mathscr{V}$ in the complete graph from the current propagation node $v$, which is readily feasible in real-world scenarios, as evidenced in the experimental findings. In a word, this property provides us with more flexibility and possibilities in adjusting the estimator parameters.

## 4.2 Proposed Algorithms

**Environmental Function.** We define the environment of node $i$ as $O_i = \mathrm{ENV}(\mathbb{H}, \boldsymbol{T}_{-i}, \boldsymbol{Z}_{-i}; )$, which summarizes the influence from its surroundings. Here, $\mathbb{H}$ denotes the network structure (e.g., hyperedges involving $i$), and $(\boldsymbol{T}_{-i}, \boldsymbol{Z}_{-i})$ represents the activation states and representation vector of all other nodes. The function $\mathrm{ENV}(\cdot)$ encodes this context into a low-dimensional vector. Since $\mathbb{H}$ is static in our setting, ENV mainly reflects the dynamic states of $i$'s neighbors, e.g., the number or embedding of active peers at a given time. Here ENV is a transmission function using Hypergraph Convolution module [3]; more details are in Appendix I.

**G-CauIM.** Our primary procedures are executed in function $Main$ (line 7). We initially train the ITE estimation model offline (line 9), represented by function $\widehat{\mathrm{ITE}}(\cdot)$ (line 1). In this model, for node $v_i$, we construct $Z_i$ via each covariate $X_i$ using representation techniques (line 2). Additionally, we construct $O_i$ to denote the higher-order interference representation of node $v_i$ with its environments: $O_i = \mathrm{ENV}(\mathbb{H}, \boldsymbol{T}_{-i}, \boldsymbol{Z}_{-i})$ (line 3), and $\boldsymbol{Z}_{-i} := \{Z_1, ...Z_{i-1}, Z_{i+1}, ...Z_n\}$. Finally, with the combination of representation $Z_i$ and $O_i$, we obtain the estimation value $\hat{Y}_i(1), \hat{Y}_i(0)$ via MLP model (line 4). During training, we employ a balancing mechanism to ensure covariate balance between the treatment group $\{v_i : T_i = 1\}$ and control group $\{v_j : T_j = 0\}$, achieved by incorporating an additional penalty term to the representation vector. Such technique is not unique, referring to Yao et al. [57], Harshaw et al. [20]. In addition, line 10 is to ensure the estimated ITE is bounded by $M$ (identified in assumption 3.1) via controlling the estimator parameter $\theta$. Diffusion process and greedy selection take place in line 11-14, where we employ the traditional greedy algorithm strategy and re-analyze the marginal benefit function as the incremental form of ITE. It is noteworthy that as the seed set expands, the activated states of each node change in the propagation, leading to diverse values of $\hat{\tau}_i$. To bound the varying $\hat{\tau}_i$, we adjust the parameters $\theta^{opt}$ of the trained $\widehat{\mathrm{ITE}}(\cdot)$, as depicted in line 12. The detailed procedure is also depicted in Fig. 5.

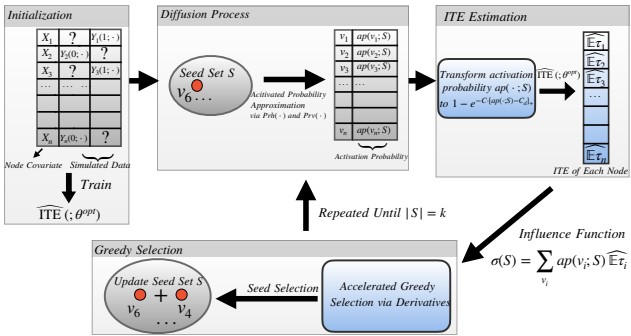

Figure 2: A-CauIM. Compared with G-CauIM (Alg. 1), we add a storage table for activation probabilities $ap(\cdot;)$ and then simplify the complex greedy selection (Eq. 5) into more efficient derivative operations (Eq. 7). In addition, we transform $ap(\cdot;)$ into continuous values close to $0, 1$ to signify the activated states $T_i$ of each node on average. And by this procedure, we obtain $\widehat{\mathbb{E}\tau_i}$ which is the approximation of the expectation on unobserved $\tau_i$.

**A-CauIM.** Four components are presented in Fig. 2. Initialization is to obtain a trained $\widehat{ITE}(;\theta^{opt})$, the same as that of G-CauIM. In the diffusion process, our objective is to calculate $ap(\cdot;)$. To address the inefficiency loops of G-CauIM, we enhance the computation by utilizing PYTORCH for fast graph computation. In sum, during each round of seed selection, we initially utilize a bipartite graph, where two sides are constructed from hyperedges and nodes, to approximately compute the activated probability $ap(v_i; S)$ of each node $v_i$ under $S$ iteratively. Specifically, during $j$-th iteration of propagation (given the current seed set $S$), according to the SICP model mentioned in Section 3, suppose that hyperedge $h_p$ is chosen, and its internal node $v_q$ is activated. The activation probability of $h_p$ and $v_q$ in $j$-th iteration are computed as $Prh(h_p, j)$ and $Prv(v_q, j)$, respectively, as defined in Eq. 6. Here $\mathcal{H}_q$ denotes the set of hyperedges containing $v_q$, $P_{SICP}$[6] represents the basic activation probability of a node. In this sense, we derive $ap(v_q; S) = \lim_{n \to +\infty} Prv(v_q, n)$. This process provides a rapid approximation of the original multiple randomizations of propagation.

$$Prh(h_p, j) = \sum_{v_k \in hp} Prv(v_k, j - 1)/|\mathcal{H}_k|,$$
$$Prv(v_q, j) = 1 - \prod_{h_p \in \mathcal{H}_q} (1 - P_{SICP})^{Prh(h_p, j)} \tag{6}$$

The next step is ITE estimation. We convert $ap(v_i; S)$ into values $1 - e^{C[ap(.;S) - C_d]_+}$ to represent the activated states $T_i$ of each node on average according to the obtained probabilities. Here $C, C_d$ are a priori constants. It aims to be close to binary treatment $0, 1$ to fit the $\widehat{ITE}$ model and maintain the differentiability. Using $ITE(;\theta^{opt})$, we determine $\widehat{\mathbb{E}\tau_i}$. For the Greedy selection process, we calculate the marginal gain using $\sigma(S) = \sum_{v_i \in \mathscr{V}} ap(v_i, S)\widehat{\mathbb{E}\tau_i}$. to approximate $\mathbb{E}\left[\sum_{v_i \in \mathscr{V}} ap(v_i, S)\tau_i\right]$. aforementioned in Section 3. Ideally, we hope A-CauIM would utilize PYTORCH to differentiate the objective function (Eq. 5) to identify the node that maximizes marginal gains. However, it is usually unreliable since the indices of nodes are discrete and not amenable to differentiation. To address this issue, we use the asymptotic approximate version of $ap(v_q; S)$ (which is "parameter-based continuous") to replace the indicator of $0, 1$ on each node. On this basis, Eq. 5 has been transformed into an operation that is continuously differentiable by PYTORCH, despite the cost of losing certain information from the unused diffusion process.

$$v := \arg\max_{v \notin S} \left\{ \frac{\partial (\sigma(S))}{\partial (ap(v; S))} * ap(v; S) \right\}. \tag{7}$$

Finally, in each round, we select the node with the highest derivative value multiplied by its activated probability as the seed node in Eq. 7, which exerts the greatest impact on $\sigma(S)$ under a small perturbation to the connection probability between $v$ and $S$ in the whole propagation.

---

[6]Here $P_{SICP}$ could be seen as the same constant for each node [56, 49] and it is easily extended to the cases where nodes are attributed to different activation probabilities.

**Time Complexity.** We have reduced the complexity of this problem in our settings from $O(KRnm)$ to $O(Km\mathbb{E}_{h\in\mathcal{H}}|h|)$, where $m,n$ are the numbers of hyperedges and nodes, $K$ is seed set number identified in our preliminaries, and $R$ is simulation number of propagation process (The complexity of the ITE estimation module, utilized in both algorithms, is excluded here). Such improvement is especially significant on relatively sparse graphs. Since $O(Km\,\mathbb{E}_{h\in\mathcal{H}}|h|) \leq O(Kmn)$ when $\mathbb{E}_{h\in\mathcal{H}}|h| \ll n$.

### 4.3 Theoretical analysis

In this section, we first prove that the traditional greedy algorithm can be naturally extended to hypergraphs and maintains the $(1 - 1/e)$ approximate guarantee. Then we demonstrate that this approximate guarantee still holds for our CauIM algorithm (Theorem. 4.4, challenge 2). In addition, we show that CauIM's performance is robust to the estimation error of ITE (Theorem 4.5, challenge 1 and challenge 3). [7]

**Proposition 4.1.** *Our CauIM problem is NP-hard.*

**Lemma 4.2** (Approximately optimal guarantee of greedy IM on hypergraph). *The greedy method on the **hypergraph** can achieve the $(1 - \frac{1}{e})$ approximate optimal guarantee.*

*Condition* 4.3 (Bounded increase of reachable probability). We define the reachable probability from a set (or node) $v_1 \in \mathcal{V}$ to a set (or node) $v_2 \in \mathcal{V}$ as $p_{v_1v_2}$. $\forall v_1 \subseteq v_2 \subseteq \mathcal{V}, |v_2| = |v_1| + 1$, we have the bounded condition of the increase of the reachable probability: $\forall v' \in \mathcal{V}, |p_{v'v_1} - p_{v'v_2}| \leq \varepsilon_1$. Moreover, $\max_{v\in\mathcal{V}} \sum_{v_i\in R(v)} |\tau_{v_i}| p_{vv_i} \leq \varepsilon_2$, where $R(v)$ denotes the successors that $v$ can arrive during the diffusion process. Here $\varepsilon_1, \varepsilon_2$ are both a priori constants.

Notice that this condition is fairly broad and model-free.

**Theorem 4.4** (Approximate optimal guarantee of CauIM). *1) If $\tau_i > 0, i \in \mathcal{V}$, the CauIM algorithm can achieve the $\left(1 - \frac{1}{e}\right)$ optimal approximate guarantee. 2) If we do not have $\tau > 0$, then a more generalized guarantee is $\sigma(S_K^g) \geq \left(1 - \frac{1}{e}\right)\left(\sigma(S^*) - K\varepsilon_1\varepsilon_2\right) - \varepsilon_2 e^{\frac{1}{K}-1}$.*

The estimation error of $\sigma(\cdot)$ can be traced back to both the Monte-Carlo strategy and the estimation error of $\tau_i$ during representation learning. We summarize it as the result on robustness analysis as follows. The proofs of the theorems are deferred in Appendix D, E, and F.

**Theorem 4.5** (Robustness). *We denote the MC estimation of $\sigma(S)$ as $\hat{\sigma(S)}$. If $\forall S \subseteq \mathcal{V}, |\frac{\hat{\sigma}(S)}{\sigma(S)} \in [1 - \gamma, 1 + \gamma]$ and $\gamma \leq \frac{\varepsilon/k}{2+\varepsilon/k}, \gamma > 0$, then our CauIM problem can achieve the optimal guarantee, which can be transferred to $\sigma(S_K^g) \geq \left(1 - \frac{1}{e} - \varepsilon\right)\left(\sigma(S^*) - K\varepsilon_1\varepsilon_2\right) - \varepsilon_2 e^{\frac{1}{K}-1}$.*

We refer readers to G for details. In addition, notice that the estimation error $\hat{\tau}_i - \tau_i$ also causes the error $\hat{\sigma}(S) - \sigma(S)$. Specifically, if we have $|\hat{\tau}_i - \tau_i| \leq \delta$, then $\hat{\sigma}(S) - \sigma(S)$ can also be bounded.

**Corollary 4.6** (Robustness of noise). *We consider the ideal CauIM case without MC strategy. The traditional IM objective function (i.e., $\tau_i = 1, \forall v_i \in \mathcal{V}$) is denoted as $\sigma_{naive}$. If $|\hat{\tau}_i - \tau_i| \leq \delta$ and $|\frac{\sigma_{naive}(S)}{\sigma(S)}| \leq \frac{\gamma}{\delta}$, then Theorem. 4.5 holds. Detailed proof can be found in Appendix H.*

## 5 Experiments

We perform experiments on four datasets and validate the findings presented in Section 4.3. We aim to answer the following three questions.

**RQ1: Effectiveness** (Theorem 4.4) When maximizing the sum of node ITE (overall individual gains), can our G-CauIM and A-CauIM outperform the traditional IM methods and maintain efficiency?
**RQ2: Robustness** (Theorem 4.5, Corollary 4.6) If ITE estimation is not accurate enough, can

---

[7] Antelmi et al. [2], Zheng et al. [65] claimed their hypergraph does not contain submodularity. However, their hypergraph is a special case (directed hypergraph), defined as $(H,t)$. $H$ is the set of nodes, and $t$ is the single tail node. Further, Gangal et al. [13] demonstrated the submodularity of the general hypergraph. [52] proposed the new weighted influence maximization problem. However, their attributes corresponding to each node are a priori assumed to be nonnegative, which differs from general ITE (which may be positive or negative). Moreover, Erkol et al. [11] stated that submodularity on the temporal network might not hold.

Table 1: RQ1: Performance comparison of four different methods on four datasets (seed number= 15). Our methods gain general improvements compared with baselines: Traditional Greedy (denoted "Baseline") and Random Selection.

| Methods | GoodReads | Contact | Email-Eu | SD-100 |
|---------|-----------|---------|----------|--------|
| Baseline | 297.56 | 68.12 | 735.28 | 138.91 |
| Random | 45.86 | 66.51 | 590.67 | 145.97 |
| G-CauIM | **330.25** | **69.53** | **804.28** | 151.59 |
| A-CauIM | 302.17 | 66.78 | 802.41 | **160.49** |

Table 2: A-CauIM presents a significant efficiency improvement compared with other competitive baselines (CPU, Torch 1.11.0). From G-CauIM to A-CauIM, the complexity reduces to $O\big(Km\mathbb{E}_{h\in\mathscr{H}}|h|\big)$ (Section 4).

| Methods | GoodReads | Contact | Email-Eu | SD-100 |
|---------|-----------|---------|----------|--------|
| Baseline | 1day | 6h40min | 22h | 3h |
| Random | 2s | 2s | 2s | 2s |
| G-CauIM | 1day02h | 7h | 23h | 3h |
| A-CauIM | 28s | 115s | 550s | 53s |

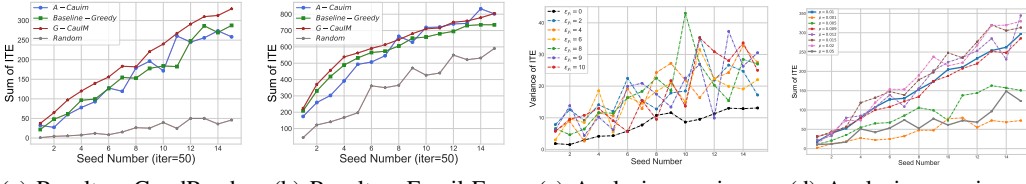

(a) Result on GoodReads    (b) Result on Email-Eu    (c) Analysis on noise $\epsilon$    (d) Analysis on various $p$

Figure 3: a,b)Performance of CauIM on the GoodReads and Contact dataset. "Iter" refers to the time step in each seed selection round. c) Variance curve trend with different noise in individual ITE, where $\epsilon_{y_i}$ represents the standard deviation of the injected noise. d) Total sum of ITE under various propagation probabilities $p$.

CauIM perform robustly? The robustness means that in perturbations, our approach will achieve an approximate result close to the normal state. **RQ3: Sensitivity** Which components and parameters of the model are essential for the performance of CauIM? Detailed Settings are outlined in Appendix C.

## 5.1 General CauIM Performance

**RQ1** Results are summarized in Table 1, with a detailed analysis of two datasets shown in Fig. 3(a) and Fig. 3(b). We can summarize the observations into four phenomena: 1) G-CauIM shows slight improvement over traditional Greedy and improves significantly compared with random selection, while the gap of their curves widens as the seed number increases. 2) A-CauIM achieves comparable performance with G-CauIM while significantly enhancing efficiency. 3) The fluctuation amplitude of the G-CauIM curve is relatively small, as demonstrated in Fig. 1, consistent with its alignment to the dynamic changes in individual effects and support from Theorem 4.4 and Theorem 4.5 in Section 4.3. 4) Traditional Greedy loses its advantage in most situations and approaches Random.

## 5.2 Robustness and Sensitivity Analysis

We address RQ2 and RQ3 using A-CauIM. It is sufficient since A-CauIM is a learning-based method with slightly more uncertainty but outstanding efficiency. We conduct experiments on GoodReads dataset for ease of analysis.

**RQ2 & RQ3** To examine robustness of our model, we add Gaussian noise $\epsilon_y$ to the simulated individual ITE results, where $\epsilon_y \sim N(0, \epsilon^2)$. The randomness of experiments thus comes from three sources: 1) $\epsilon_y$, 2) the propagation probability $p$, and 3) the dynamic ITE of each node. We modify the scale of the noise and plot corresponding curves. Fig. 3(c) can provide the following illustrations: 1) when the noise increases (not exceeding 9), it is not enough to counter the random variance of the propagation process and dynamic ITE itself. This "powerlessness" effect disappears until noise reaches around 10. 2) During the early stage of seed growth, when the number of seeds is smaller than 10, the effect of total noise of the infected nodes is not large. More discussions of the parameter $\epsilon$ are in Appendix C.5.

We further detect how the components of the algorithm interact and how the diffusion process affects A-CauIM with various parameter $p$; As shown in Fig. 3(d), curves of total ITE nearly merge to one when $p$ is between 0.01 and 0.05. This convergence indicates that complete traversal of nodes in the hypergraph occurs with a sufficiently high propagation probability, leading to the stabilization

of the total dynamic ITE. Additionally, it is noteworthy that the performance sharply declines when $p = 0.05$, possibly due to the broader diffusion process intensifying the randomness in dynamic ITE and consequently augmenting the uncertainty of influence spreading. In conclusion, with p changing in a certain range, our algorithm remains stable, and $p = 0.02$ is an empirically good choice for achieving the best performance.

## 6 Conclusion and Discussion

In this paper, we analyze traditional IM from a causality perspective. Our CauIM framework can extract approximately optimal seed sets to achieve novel influence maximization. *Discussion on Sketch-based Models.* While sketch-based models are efficient, they may struggle to adapt to dynamic node weights during seed selection, unlike greedy-based approaches. This limitation can lead to instability, especially in sparse or heterogeneous networks. Our acceleration technique is flexible across different graph structures and offers a potential remedy. Future work will explore extending sketch-based approximations to more complex heterogeneous networks.

## Acknowledgements

Zhiheng Zhang is supported by "the Fundamental Research Funds for the Central Universities" (number: 2025110602) of Shanghai University of Finance and Economics. This work is also partially supported by the Basic Research Fund of the Computer Network Information Center, Chinese Academy of Sciences (Frontier Technology Innovation Project, No. E5553601).

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

Figure 4: An illustration of the propagation process. With $v_6$ infected, the hyperedges ($h_2$, $h_3$ and $h_4$ with solid line) containing $v_6$ are chosen to be the candidates. Nodes ($v_3, v_4, v_5, v_7$) in these hyperedges will potentially convert to the infected state. The spreading probability $P_{IC}$ is the internal parameter in the hypergraph-based Independent Cascading (IC) model [56].

# A  Methodology Details

Further methodological details, extended analyses, additional experiments, and core codes are provided in https://github.com/suxinyan/cauim3236/.

Table 3: Notations.

| Symbol | Descriptions |
| --- | --- |
| $v_i, h_j$ | node $i$ ,hyperedge $j$ in hypergraph $\mathcal{G}$: $v_i \in \mathcal{V}$, $h_j \in \mathcal{H}$ |
| $\mathbb{H}$ | the hypergraph structure matrix |
| $X_i$ | the covariate of node $v_i$ |
| $T_i$ | the activating status (treatment) of node $v_i$ |
| $\boldsymbol{X}_{-i}, \boldsymbol{T}_{-i}$ | environmental information of node $v_i$, including covariates and the activation status of surrounding nodes (see Eq. 1) |
| $\hat{Y}_i(1;\cdot), \hat{Y}_i(0;\cdot)$ | the estimated potential outcome of node $v_i$ |
| $\hat{\tau}_i$ | the estimated ITE of node $v_i$ in certain diffusion process through Function $\widehat{\text{ITE}}(\cdot)$ (see Alg. 1) |
| $ap(v_i, S)$ | diffusion probability from set $S$ to the node $v_i$ |
| $\widehat{\mathbb{E}\tau_i}$ | approximation of expectation on $\tau_i$ |

The procedure of G-CauIM is provided in Fig. 5.

# B  Comparison

A detailed comparison of our CauIM within three types of IM frameworks is presented in Table 4, while a comparison of CauIM with the general traditional IM framework is provided in Table 5. As shown in Table 4, the Simulation-based CauIM is successfully implemented in this work.

# C  Experiments Details

## C.1  Experimental Settings

Our real-world data comprises three real-world public datasets: GoodReads [8], Contact [39], and Email-Eu [4]. Furthermore, we incorporate a synthetic dataset named SD-100 comprising 100 nodes and 100 hyperedges, whose initial treatments and feature settings are detailed in Section C.2. We compare the performance of G-CauIM and A-CauIM with the traditional greedy selection method without parameter adjusting strategy (Noted as "Baseline") on the aforementioned four datasets. Additionally, we choose the randomized seed selection strategy as another baseline. We randomly conduct each experiment for 10 times and each time for 20 rounds in influence estimation. For basic hyperparameters, we set seed number $K = 15$ and spread probability $P_{SICP}$ as 0.01 ( denoted as $p$ for simplicity). The evaluation metric is the sum of ITE spread by selected seeds with the same trained ITE estimation module illustrated in Section 4.

---

[8] https://www.goodreads.com/

| | Simulation-based | Proxy-based | Sketch-based |
|---|---|---|---|
| Basic idea | Use Monte-Carlo (MC) simulation to evaluate ITE influence spread | Design proxy models to approximate influence function with varying ITE | Construct reachability/activation sketches to accelerate greedy selection; guarantees typically assume fixed, context-independent, nonnegative gains (can be used as auxiliary approximations within CauIM when negative gains are limited or controlled.) |
| property | NP-hard complexity, total theoretical guarantee | Polynomial/linear complexity, no theoretical guarantee | Quasi-linear complexity, approximation guarantees under fixed nonnegative gains and specific diffusion models; no guarantee for context-dependent or negative ITE |
| Disadvantages | Computational overheads | Sensitive to the unstable scenarios | Not general to a wider range of diffusion models |
| Datasets | Small to medium-sized datasets under all propagation models | Large-scale datasets with distinctive graph structure under specific propagation models | Large graphs under Triggering/LT settings with fixed gains; less suitable when gains are dynamic |
| Examples | Greedy, CELF (famous) | HADP [56], HSD [56], EIOA [41] | RIS [5], IMM [49], BKRIS |

Table 4: Comparison of three types of traditional IM in the context of our CauIM framework.

| | CauIM | Traditonal IM |
|---|---|---|
| Basic idea | Leverages observational data to estimate the ITE of each node and to maximize the sum of varying ITEs among the infected individuals considering environmental information | Maximize the numbers of infected individuals |
| Objective function | $\arg\max_{S \subseteq \mathcal{V} \wedge |S|=K} \mathbb{E}[|\Phi(S)|]$, where $\Phi(S) = \sum_{v_i \in S} \mathbb{E}\tau_i$ measures the sum of ITE (informal, parameter of interference is omitted) | $\arg\max_{S \subseteq \mathcal{V} \wedge |S|=K} \mathbb{E}[|\Phi(S)|]$, where $\Phi(S)$ measures the sum of infected numbers |
| Application Scenarios | Maximize total sum of individual gains | Maximize infected numbers |

Table 5: A comparison between CauIM and general traditional IM (Notice that CauIM is more challenging than the sum-weighted IM, since the ITE would be negative and varying during different propagation.)

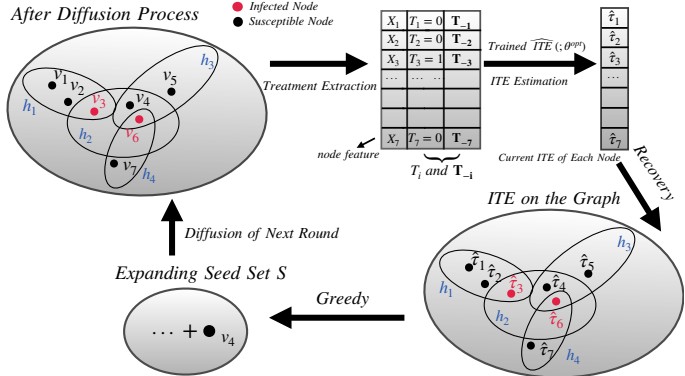

Figure 5: The procedure of G-CauIM. $T_i$ indicates whether the node is activated or not (we can only observe one situation for each node $v_i$) and $\boldsymbol{T_{-i}}$ represents activated status of its surroundings, as illustrated in Eq. 2. For each round, we construct the ITE estimation $\hat{\tau}_i := \widehat{ITE}(;\theta^{opt})$ mentioned in Section 4 and then treat it as the node weight. Furthermore, we conduct a weighted greedy algorithm with SICP propagation mechanism (Fig. 4). As a result, we expand the seed set ($v_4$ is added), targeting the (estimated) largest sum of ITE. The main challenge is that $\hat{\tau}_i$ are not constants, since the omitted parameter (Eq. 2) changes according to the different activation status of nodes in each iteration.

## C.2  Details of Datasets and Problem Background

GoodReads [22, 50] collects information on different categories of books, with each item containing the book title, content, and other details. Using the "Author-book" relationship, a node represents the specific book category and hyperedge aggregates books written by the same author in our hypergraph. We consider a scenario of recommending book sales, where the diffusion process is facilitated by reader groups associated with each book category node in the "Author-book" network. Treatment denotes the recommendation for book sales, with $t_i$ set to 1 when book node $i$ is recommended. Each book category is associated with a potential sales income, influenced by both its own recommendation status and the performance of other books within the same hyperedge. Our goal is to identify a k-set(or k kinds) of books to sell at the beginning, aimed at maximizing the total sales gap between the recommended and non-recommended. Our core optimization function is to maximize the sum of ITE, where the ITE of each book means the difference in potential sold income with/without the recommendation in our experiment. For Contact [4], it constructs simplicial complexes by grouping individuals in close connectivity at the same timestamp, represented by hyperedges. Here we simulate a situation where we deliver an AIDS Awareness Talks to particular students, and the core concepts can be disseminated through hyperegde groups. Our objective is to select initial student representatives to be educated in order to maximize the overall benefits of the talk (This can also be viewed as maximizing the total sum difference between having the anti-drug talk and not having it). We simulate covariates($X_i$) of each student in a Mixed Gaussian distribution, considering differences among diverse groups:

$$X_i \sim \sum_{j=1}^{L} \omega_i \mathcal{N}(\mu_j, I). \tag{8}$$

Here we set $L = 4, \omega_1 = 0.4$. Moreover, $\{\omega_1, \omega_2, \omega_3, \omega_4\}= \{0.4, 0.2, 0.1, 0.3\}$, $\{\mu_1, \mu_2, \mu_3, \mu_4\} = \{0.2, -0.25, -0.3, 0.5\}$.

Email dataset shares the similar scenario with Contact. The ratio of nodes to hyperedges is different: For example, Goodreads is 36 and Contact is 0.14. The simulation of ITE and basic treatment settings for both datasets follow [37]. The ITE distribution of GoodReads and Contact is shown in Fig. 6. It is worth noting that the ratio of nodes to hyperedges is contrasting between the two datasets, yielding sparse and relatively dense graphs, respectively. This exemplifies the diversity of our data selection, allowing for a more robust evaluation of algorithmic performance. Synthetic dataset SD generates initial treatments $T_i \sim \text{Bernoulli}(r_0)$, where $r_0$ denotes an average affected ratio of nodes (which can be calculated easily through propagation simulations). Its covariates($X_i$) are simulated following Eq. (8), with distinct values of $\{\omega_1, \omega_2, \omega_3, \omega_4\} = \{0.4, 0.25, 0.15, 0.2\}$, representing different group ratio.

## C.3  Supplementary Descriptions of Basic Assumptions

The basic assumptions of the book-selling scenario should be satisfied: 1) Books and authors are many-to-many relationships; 2) the number of each kind of books sold initially is the same (we will take it as our future research

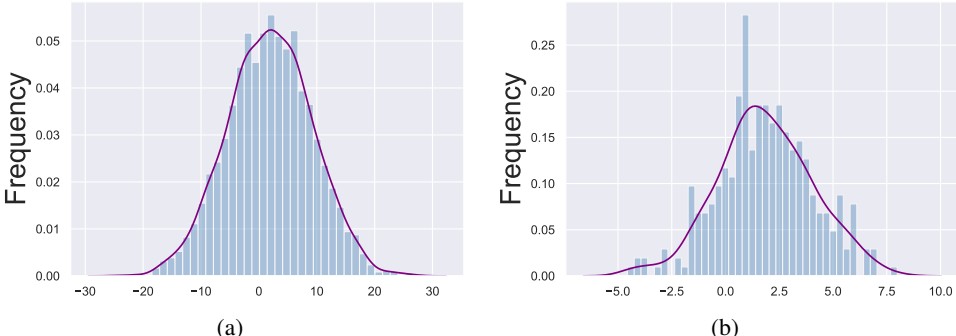

Figure 6: a) Distribution of individual ITE in GoodReads Dataset. b) Distribution of individual ITE in Contact Dataset.

topic if not the same); 3) Readers can learn about other books simply from authors of owned books. 4) Temporal change of each hyperedge is not considered. Assumptions for the other example are similar with the exception that rule 3 is replaced by "Students can learn about core ideas simply from other students." Our diffusion framework is established by hypergraph-based IC as described in Section 4. Within this model, the spread probability represents the willingness of readers to purchase the next book in the GoodReads dataset and the likelihood of core concepts spreading among students in the Contact or Email dataset.

### C.4 Supplementary Descriptions of Parameter Settings

Our experiments are conducted on Linux operating system with Python 3.10.14, torch 2.1.

### C.5 Parameter Analysis

Still, after the number of seeds exceeds 10, the scale of affected nodes rises significantly, since the cumulative effect of noise becomes significant. Thus, we conclude that our model remains relatively robust when the noise standard deviation $\sigma$ does not exceed 8. This result is consistent with our theoretical part. Aimed at indicating the volatility, we modify the scale of the noise to approximate instability degree and take on 10 realizations of each experiment, calculating the standard variance of ITE performance among those realizations as the final result. We conduct 20 groups of experiments with $\sigma$ varying between 0 to 20, and use step 2 when $\sigma$ is lower than 8 for its changes are not noticeable. While $\sigma$ is larger than 10, the curves increase too drastically. Thus, results beyond this range are omitted from Fig. 3(c) for clarity.

## D    The proof of Proposition 4.1

Due to the Influence Maximization (IM) problem itself being NP-hard, our CauIM can be naturally reduced to the traditional IM problem ($\tau = 1$) and is therefore also NP-hard, potentially involving additional complexity due to heterogeneous node weights.

*Proof sketch.* Notice that when $\tau_i = 1, \forall v_i \in \mathscr{V}$ and each hyperedge contains only one pair of nodes, then CauIM will degenerate to the traditional IM problem. Moreover, the IM problem with the IC diffusion model has been demonstrated to be NP-hard in [28].

In another perspective, we can directly prove the optimal seed set of CauIM is one solution of the famous weighted set cover problem, which is well-known as NP-hard [21]. The weighted set cover problem is equivalently defined as detecting whether there exists $k$ subsets within the total $m$ subsets, such that it can cover the universe of elements $U$. We aim to demonstrate the traditional weighted set cover problem can naturally generalize a specific instance in CauIM. We construct a bipartite graph, in which the left side denotes the total subsets $\mathscr{S} = \{\mathcal{S}_1, \mathcal{S}_2, ...\mathcal{S}_m\}$, while the right side denotes the elements $u \in U$, which can be seen as each node in the hypergraph. The edge between two sides is attributed with the probability 1. Notice that in this instance, there exists a $k$-set cover if and only if there exists $k$-seed set such that the activation can reach all $|S| + |U| = K + |U|$ nodes. Hence, our CauIM is more complex and is NP-hard.

$\square$

# E   The proof of Lemma 4.2

*Proof.* It is equivalent to consider CauIM with $\tau_i = 1, v_i \in \mathcal{V}$. This approximate optimal guarantee is due to two elegant properties: 1) monotonicity and 2) submodularity. Firstly, according to $\sigma(S_0 \cup v) - \sigma(S_0) \geq 0$ when $\tau_i = 1$, the monotonicity naturally holds. Secondly, we consider the submodularity in the hypergraph. This part has been proved by [13], where they constructed an augmented graph $G^{aug} = (\mathcal{V} \cup \mathcal{H}, \mathcal{E})$. Here the edge $\mathcal{E}$ is composed by $e := (v, h), v \in \mathcal{V}, h \in \mathcal{H}$. Then conduct the same submodularity analysis as in the traditional graph, and the proof is completed.

$\square$

# F   The proof of Theorem 4.4

The first result based on $\tau_i > 0$ naturally holds. It is because the monotonicity and submodularity still hold when each node is attributed with non-negative weights (*i.e.*, ITE). We focus on the second result, when $\tau_i > 0$ is not guaranteed, these two important properties will not further hold. We extend the analysis to a generalized form of weak monotonicity and weak submodularity.

*Proof.* To summarize, the core part is the following three claims:

**(Claim 1)** $\sigma(S^*) \leq \sigma(S^* \cup S_i^g) + i\varepsilon_2$.

**(Claim 2)** $\sigma(S^* \cup S_i^g) \leq \sigma\left(S_{i+1}^g\right) - \sigma\left(S_i^g\right) + \sigma\left(S_i^g \cup S_{K-1}^*\right) + \varepsilon_1\varepsilon_2$.

**(Claim 3)** $\sigma\left(S_i^g \cup S_{K-1}^*\right) \leq (K-1)[\sigma\left(S_{i+1}^g\right) - \sigma\left(S_i^g\right)] + \sigma(S_g^i) + (K-1)\varepsilon_1\varepsilon_2$. The optimal $K$-seed set is denoted as $S^* = \{s_1^*, s_2^*, ...s_K^*\}, S_k^* = \{s_1^*, s_2^*, ...s_k^*\}$, and the set output from our greedy CauIM as $S^g = \{s_1^g, s_2^g, ...s_K^g\}, S_k^g = \{s_1^g, s_2^g, ...s_k^g\}, k \in [K]$. Following [52], notice that

$$\sigma(S) = \sum_{u \in S} \left( \sum_{v \in V} \tau_v \cdot p_r(u, v) + \tau_u \right). \tag{9}$$

We first construct the facilitating claim to analyze the variant of the monotonicity and the submodularity property.

**Claim 1:** $\sigma(S^*) \leq \sigma(S^* \cup S_i^g) + i\varepsilon_2$.

This claim can be achieved recursively. Considering two sets $T_1 \subseteq T_2$, and an additional vertex $v \nsubseteq T_1$, we can follow [52] and achieve:

$$(\sigma(T_1 \cup v) - \sigma(T_1)) = \sum_{v_i \in R(v)} \tau_i \cdot p_{vv_i} (1 - p_{T_1, v_i}) \leq \sum_{v_i \in R(v)} |\tau_i| p_{vv_i} = \varepsilon_2. \tag{10}$$

Then Claim 1 follows by recursively applying this bound $i$ times.

**Claim 2:** $\sigma(S^* \cup s_i^g) \leq \sigma\left(S_{i+1}^g\right) - \sigma\left(S_i^g\right) + \sigma\left(S_i^g \cup S_{K-1}^*\right) + \varepsilon_1\varepsilon_2$.

We make an extension of submodularity. Considering $S \subseteq T \subseteq \mathcal{V}$, we have

$$(\sigma(S \cup v) - \sigma(S)) - (\sigma(T \cup v) - \sigma(T))$$
$$= \sum_{v_i \in R(v)} \tau_i \cdot p_{vv_i} (p_{T, v_i} - p_{S, v_i}) \leq \varepsilon_1\varepsilon_2. \tag{11}$$

Hence $\sigma(S^* \cup s_i^g)$ can be bounded as follows:

$$\sigma(S^* \cup s_i^g) = \sigma(S_{K-1}^* \cup s_K^* \cup S_i^g)$$
$$\leq \sigma\left(S_i^g \cup s_K^*\right) - \sigma\left(S_i^g\right) + \sigma\left(S_i^g \cup S_{K-1}^*\right) + \varepsilon_1\varepsilon_2 \tag{12}$$
$$\leq \sigma\left(S_{i+1}^g\right) - \sigma\left(S_i^g\right) + \sigma\left(S_i^g \cup S_{K-1}^*\right) + \varepsilon_1\varepsilon_2.$$

The last line is due to the selection nature of the greedy algorithm.

**Claim 3:** $\sigma\left(S_i^g \cup S_{K-1}^*\right) \leq (K-1)[\sigma\left(S_{i+1}^g\right) - \sigma\left(S_i^g\right)] + \sigma(S_g^i) + (K-1)\varepsilon_1\varepsilon_2$.

It is due to

$$\sigma\left(S_i^g \cup S_{k-1}^*\right) = \sigma\left(S_i^g \cup S_{k-2}^* \cup s_{k-1}^*\right)$$
$$\leq \sigma\left(S_i^g \cup s_{k-1}^*\right) - \sigma\left(S_i^g\right) + \sigma\left(S_g^i \cup S_{k-2}^*\right) + \varepsilon_1\varepsilon_2 \tag{13}$$
$$\leq \sigma\left(S_{i+1}^g\right) - \sigma\left(S_i^g\right) + \sigma\left(S_g^i \cup S_{k-2}^*\right) + \varepsilon_1\varepsilon_2$$

The last line is due to the selection nature of the greedy algorithm. Therefore, recursively, we have

$$\sigma\left(S_i^g \cup S_{K-1}^*\right) \leq (K-1)[\sigma\left(S_{i+1}^g\right) - \sigma\left(S_i^g\right)] + \sigma(S_g^i) + (K-1)\varepsilon_1\varepsilon_2. \tag{14}$$

Combined with Eq. (12), Eq. (F) and **Claim 3**, we have

$$\sigma(S^*) \leq K[\sigma\left(S_{i+1}^g\right) - \sigma\left(S_i^g\right)] + \sigma(S_i^g) + K\varepsilon_1\varepsilon_2 + i\varepsilon_2. \tag{15}$$

It equals to

$$\sigma(S_{i+1}^g) \geq (1 - \frac{1}{K})\sigma(S_i^g) + \frac{\sigma(S^*)}{K} - \varepsilon_1\varepsilon_2 - \frac{i\varepsilon_2}{K}$$

$$(1 - \frac{1}{K})^{k-i-1}\sigma(S_{i+1}^g) \tag{16}$$

$$\geq (1 - \frac{1}{K})^{k-i}\sigma(S_i^g) + (1 - \frac{1}{K})^{k-i-1}\left(\frac{\sigma(S^*)}{K} - \varepsilon_1\varepsilon_2 - \frac{i\varepsilon_2}{K}\right).$$

Take the sum of $i \in \{0, 2, ...K - 1\}$, we have

$$\sigma(S_K^g) \geq \sum_{i=0}^{K-1}(1 - \frac{1}{K})^{k-i-1}\frac{\sigma(S^*) - K\varepsilon_1\varepsilon_2}{K} - \sum_{i=0}^{K-1}(1 - \frac{1}{K})^{k-i-1}\frac{i}{K}\varepsilon_2$$

$$\geq \left(1 - \left(1 - \frac{1}{k}\right)^k\right)(\sigma(S^*) - K\varepsilon_1\varepsilon_2) - \varepsilon_2(\frac{1}{e})^{1-\frac{1}{K}}\int_0^1 xe^x dx \tag{17}$$

$$\geq (1 - \frac{1}{e})(\sigma(S^*) - K\varepsilon_1\varepsilon_2) - \varepsilon_2 e^{\frac{1}{K}-1}.$$

This lower bound is also applicable when $\tau_i \leq 0$ exists.

$\square$

# G    The proof of Theorem 4.5

*Proof.* Following Chen et al. [8] , we have ($k \in [K]$)

$$f\left(S_i^g \cup \{s_k^*\}\right) \leq \frac{1}{1-\gamma}\hat{f}\left(S_i^g \cup \{\bar{s}_k^*\}\right)$$

$$\leq \frac{1}{1-\gamma}\hat{f}\left(S_i^g \cup \{s_{i+1}\}\right) \tag{18}$$

$$\leq \frac{1+\gamma}{1-\gamma}f\left(S_i^g \cup \{s_{i+1}\}\right).$$

Analogously, we have

$$f\left(S_{i+1}^g\right) \geq \frac{1-\gamma}{1+\gamma}\left(\left(1 - \frac{1}{K}\right)f\left(S_i^g\right) + \frac{f\left(S^*\right)}{K} - \varepsilon_1\varepsilon_2 - \frac{i\varepsilon_2}{K}\right). \tag{19}$$

**Algorithm 2:** Monte Carlo-based greedy CauIM

**Require:** Hypergraph $\mathscr{G}(\mathscr{V}, \mathscr{H})$, size of the seed set $K$, a constant $T$.
1: Initialization: $S_0 = \emptyset$, $k = 0$.
2: **ITE recovery:** Estimate node-level $\tau_i$ from observational data or model-based inference.
3: **for** $|S_0| < K$ **do**
4:     $v_0 = \arg\max_{v \notin S_0} \{ \mathbf{MC}(S_0 \cup \{v\}, T) - \mathbf{MC}(S_0, T) \}$.
5:     $S_0 = S_0 \cup \{v_0\}$
6: **end for**
**Ensure:** The deterministic seed set $S_0$ with $|S_0| = K$.
    **Function MC:**
**Require:** Iteration $T$, current node set $S_0$, estimated ITE $\{\tau_i\}$
7: count $= 0$
8: **for** $i \in [T]$ **do**
9:     We conduct the diffusion process with $T$ steps, and compute the sum of causal effects
      $\sigma_T(S_0) = \sum_{\text{node } j \text{ is activated}} \tau_j$; count $=$ count $+ \sigma_T(S_0)$.
10: **end for**
**Ensure:** Return $count/T$.

Hence, recursively,

$$
\begin{aligned}
f(S_K^g) &\geq \sum_{i=0}^{K-1} \left( \frac{(1-1/K)(1-\gamma)}{1+\gamma} \right)^{K-i-1} \cdot \frac{1-\gamma}{(1+\gamma)K} \cdot [f(S^*) - K\varepsilon_1\varepsilon_2] \\
&\quad - \sum_{i=0}^{K-1} \left( \frac{(1-1/K)(1-\gamma)}{1+\gamma} \right)^{k-i-1} \frac{i}{K} \varepsilon_2 \\
&\geq \frac{1 - \left(\frac{1-\gamma}{1+\gamma}\right)^K \left(1 - \frac{1}{K}\right)^K}{(1+\gamma)K/(1-\gamma) - K + 1} [f(S^*) - K\varepsilon_1\varepsilon_2] - \varepsilon_2 e^{\frac{1}{K}-1} \\
&\geq \frac{1 - \left(\frac{1-\gamma}{1+\gamma}\right)^K \cdot \frac{1}{e}}{(1+\gamma)K/(1-\gamma) - K + 1} [f(S^*) - K\varepsilon_1\varepsilon_2] - \varepsilon_2 e^{\frac{1}{K}-1} \\
&\geq \frac{1 - \frac{1}{e}}{(1+\gamma)K/(1-\gamma) - K + 1} [f(S^*) - K\varepsilon_1\varepsilon_2] - \varepsilon_2 e^{\frac{1}{K}-1} \\
&\geq \left(1 - \frac{1}{e}\right) \left(1 - \frac{(1+\gamma)K}{1-\gamma} + K\right) [f(S^*) - K\varepsilon_1\varepsilon_2] - \varepsilon_2 e^{\frac{1}{K}-1} \\
&\geq \left(1 - \frac{1}{e} - \left( \frac{(1+\gamma)K}{1-\gamma} - K \right)\right) [f(S^*) - K\varepsilon_1\varepsilon_2] - \varepsilon_2 e^{\frac{1}{K}-1} \\
&\geq \left(1 - \frac{1}{e} - \varepsilon\right) [f(S^*) - K\varepsilon_1\varepsilon_2] - \varepsilon_2 e^{\frac{1}{K}-1}.
\end{aligned}
\tag{20}
$$

$\square$

## H   The proof of Corollary 4.6

*Proof.*

$$
\begin{aligned}
|\hat{\sigma}(S) - \sigma(S)| &= \sum_{u \in S} \left( \sum_{v \in V} (\hat{\tau}_v - \tau_v) p_r(u,v) + \hat{\tau}_v - \tau_v \right) \\
&\leq \frac{\gamma\sigma(S)}{\sigma_{naive}(S)} \sum_{u \in S} \left( \sum_{v \in V} p_r(u,v) + 1 \right) := \gamma\sigma(S).
\end{aligned}
\tag{21}
$$

Hence $|\frac{\hat{\sigma}(S)}{\sigma(S)} - 1| \leq \delta \frac{\sigma_{naive}(S)}{\sigma(S)} \leq \gamma$.        $\square$

# I   Auxiliary algorithms and Additional discussions

**More discussions of HGCN module**   The higher-order interference representation $O_i()$ is learned according to to [37, 38], employing hypergraph convolution operator within HGCN module: $\mathbf{O}^{(l+1)} = \text{LeakyReLU}\left(\mathbf{L}\mathbf{O}^{(l)}\mathbf{W}^{(l+1)}\right)$ where $\mathbf{L}$ denotes Laplacian matrix aggregating the graph feature information, and $\mathbf{O}$ is initially calculated using $Z$ .

**More discussions upon IM**   For the *first* limitation of IM, the exploration of hypergraph-based IM is urgent to be settled. Since the hypergraph structure is consistent with ample real-world scenarios, especially when different nodes in the graph contain high-level, multivariate relationships, where the traditional graph is hard to model efficiently. Take an example of the disease propagation problem regarded as IM on hypergraph in Fig. 4(a). Students are connected through social circles, where each circle can be represented as a hyperedge. Different from ordinary graphs, the influence of node $v_6$ is spread not considering the edge consisting of a pair of nodes, but on more its affiliated hyperedges shown in Fig. 4(b). Existing hypergraph-based studies are mostly separated into two parts: 1) people are committed to developing heuristic methods but with not enough theoretical support [2, 56]. 2) people developed fundamental theoretical guarantees only on a specific form of hypergraph structure [66]. In general, a general hypergraph-based IM with theoretical guarantee and high empirical efficiency still needs to be explored.

For the *second* limitation of IM, the original optimization objective needs to be reconsidered in many cases. The previous objective is directly implied as the sum of node numbers, which stems from empirical or even philosophical determinations and lacks rigorous mathematical arguments. This implication is attributed to the over-simplification of real-life situations–current IM methods tend to overlook the dynamic nature of node influence weights (ITE) within their environments. Traditional IM methods like simulation-based [28, 31] and sketch-based [5, 48, 49, 51] ones focusing on maximizing total numbers (or generalized weighted IM) might fail to pursue such maximum total potential benefit (pursue "larger varying weighted sum" instead of "larger number of nodes"). Wang et al. [52] proposed the new weighted IM problem whereas they rely on non-negative assumptions and and lacks generalizability to complex scenarios involving hypergraphs and varying node weights. Recently, learning-based IM methods [6, 27, 30, 42, 35] mostly learn potential node representations as a marginal gain of node influence, thus guiding the seed node finding process. Sharing different object functions from ours, many of these existing methods might struggle with limited generalization capabilities and result reliability concerns. Overall, there is a critical need to explore novel objective functions.

**More discussions upon extended algorithms**   To simplify the discussion, in the additional algorithms we provide, we have omitted the process of dynamically updating ITE based on the surrounding nodes' state changes during each propagation. CELF-CauIM (Alg. 2) and Monte Carlo-based greedy CauIM (Alg. 3) can be derived naturally from G-CauIM.

---

**Algorithm 3:**  CELF-CauIM

**Require:** Hypergraph $\mathcal{G}(\mathcal{V}, \mathcal{H}, \mathbb{H})$, size of the seed set $K$, causal influence function $\sigma(\cdot)$.
 1: Initialization: $S^* = \emptyset$, $MargDic = \emptyset$.
 2: **ITE recovery**.
 3: $MargDic$ stores the marginal gain $\sigma(\{v\})$ of each node.
 4: Sort $MargDic$ in decreasing order of value.
 5: **for** $|S^*| < K$ **do**
 6:     Move out the node $cur$ with the largest marginal gain in $MargDic$.
 7:     Re-compute marginal gain of of node $cur$ with the current seed set $S^*$:
     $MargDic[cur] = \sigma\left(S^* \cup \{cur\}\right) - \sigma\left(S^*\right)$
 8:     Check if previous top node stays on top after sort $MargDic$ again. If true, $S^* = S^* \cup \{cur\}$, and find the second seed; else remove the second largest marginal gain node in $MargDic$, then repeat the last operation.
 9: **end for**
**Ensure:** The deterministic seed set $S^*$ with $|S^*| = K$.

---

For HADP-CauIM (Alg. 4), we replace the selection criterion of searching for nodes with the highest degree with the nodes with the highest sum of the average ITE among the neighboring nodes (line $5 - 6$). Note that according to the common drawback of heuristic methods, this type of method has no theoretical support and often falls into a common dilemma: nodes with the highest local degree (neighboring ITE) may not necessarily represent seeds that can bring greater overall influence. This point has also been verified in the experiments.

---

**Algorithm 4:** HADP-CauIM

---

**Require:** $\mathcal{G}(\mathcal{V}, \mathcal{H}, \mathbb{H})$, size of the seed set $K$, causal influence function $\sigma(\cdot)$.

1: Initialization: $S^* = \emptyset$, $DegITE = \{\}$.
  /* Presented in Algorithm 1 */
2: **ITE recovery**.
3: $DegITE[v]$ stores the sum of ITE of the neighbour nodes of each node $v$, where $N_r(v)$
  represents the neighbour nodes of $v$: $DegITE[v] = \sum_{v_r \in N_r(v)} \mathbb{E}\tau_r$.
4: **while** $|S^*| < K$ **do**
5:   Choose $v_0$ with the max value in $DegITE$ as the seed: $v_0 = argmax_v\{DegITE[v]\}$,
    $S^* = S^* \cup \{v_0\}$.
6:   Calculate sum of ITE for each node $v_r$ in neighbors of $v_0$ as edge value $Edge$:
    $DegITE[v_r] = \sum_{v_{rq} \in N_r(v_r)} \mathbb{E}\tau_{rq}$ .
7:   Remove the edge influence of the chosen seed node $v_0$:
    $DegITE[v_r] = DegITE[v_r] - Edge$ .
8:   Remove $v_0$ and its incident hyperedges from $\mathcal{G}$.
9: **end while**
**Ensure:** The deterministic seed set $S^*$ with $|S^*| = K$.

---

**More discussions upon submodularity of hypergraph**   Antelmi et al. [2], Zheng et al. [65] claimed their hypergraph does not contain submodularity. However, their propagation mechanism is different from traditional IC. Besides, they considered the form of directed hypergraph where a hyperedge $(H, t)$ comprises a set of head nodes $H$ is and a single tail node $t$. Further, Gangal et al. [13] demonstrated the submodularity of a general class of hypergraph. Moreover, Erkol et al. [11] stated that the submodularity on the temporal network might not be held.

**More discussions upon additional challenges compared to other sum-weighted IMs.**   Noteworthy, according to the ITE estimation form, CauIM can be seen as the generalized case of the weighted IM. However, our task is significantly more challenging. Firstly, beginning with the traditional graph, sum-weighted schemes often ensure an approximate optimum guarantee effortlessly, for traditional IM can be extended to weighted IM naturally (see definition 5 in Mossel and Roch [40]). However, it does not make sense in our setting since the ITEs for each node would not guarantee to be always non-negative (for instance, some non-compliers, i.e., ITE $< 0$, exist). Furthermore, the argument on submodularity is more complex in hypergraph (defer to I). Besides, the ITE estimation does not remain constant between iterations due to the experimental sensitivity (the activation status of surrounding nodes change). In sum, to the best of our knowledge, in the setting of hypergraph-based ITE, the weakened version of the approximate optimum guarantee is an effective supplement to the IM community.

**More future work**   This study introduces a causal influence maximization (CauIM) framework that captures environmental sensitivity and unobservability in network diffusion, establishing a bridge between causal inference and influence maximization. A promising direction for future work is to further unify this causal perspective with sequential and online design paradigms developed in recent research on networked experimentation [61, 54]. Specifically, integrating the CauIM framework with online experimental design could enable adaptive intervention strategies that dynamically estimate and optimize individual influence gains while providing anytime-valid inference guarantees. Moreover, the interplay between partial identification [60, 59] and network diffusion can be explored to quantify uncertainty when the activation mechanisms or environmental variables are only partially observed, leading to bounds on causal influence under incomplete network information. Robust and proxy-based identification methods [63] may further improve reliability by mitigating hidden confounding in heterogeneous or noisy propagation environments. Another direction is to extend CauIM to dynamic or temporal networks, leveraging Granger-style causality models [62] to capture time-varying dependencies in diffusion processes. Finally, combining CauIM with active treatment-effect estimation under limited sampling budgets [64] and structural constraints such as topological regularity [58] can yield a comprehensive framework for optimizing both causal inference accuracy and influence efficiency across evolving network settings.

## J   Broader Impact.

Our work studies algorithmic optimization on synthetic and publicly available network data. Potential positive impacts include improved targeting for public-health messaging and resource allocation. Potential risks include misuse for manipulative advertising. We discuss mitigation by enforcing transparency about the optimization objective, prohibiting protected-attribute targeting, and auditing estimated ITEs for bias.

