# OpenReview forum: "Unveiling Environmental Sensitivity of Individual Gains in Influence Maximization"
_NeurIPS.cc/2025/Conference — NeurIPS 2025 poster_

### Official Review · Reviewer_ioD7 · 2025-06-21

**Clarity:** 3
**Significance:** 2
**Originality:** 2
**Rating:** 4
**Confidence:** 4

**Summary:**

This manuscript investigates influence maximization (IM) algorithms in the setting where individuals receive inconstant gains throughout the diffusion process. To address the challenges arising from unobservability and environmental sensitivity, the authors propose two algorithms, CauIM and A-CauIM. They validate their methods using both synthetic and real-world networks, demonstrating the algorithms’ effectiveness.

**Questions:**

1.	What precisely does the ‘environment information’ T_{-i} in Eq. 1 represent? Does it refer to the states of neighboring nodes or to the external context?
2.	How do the authors guarantee that the proposed algorithms converge to an optimal solution? Can the resulting node sets truly be regarded as those with maximal influence?
3.	The datasets used are relatively small in size, and the simulations are limited in the number of trials. This raises concerns about the generalizability and robustness of the proposed methods.
4.	The baseline comparison is conducted using a basic greedy algorithm, which is somewhat outdated relative to current IM literature. Given the maturity of IM research, it is not clear how the proposed methods improve upon or outperform existing state-of-the-art techniques.

**Ethical Concerns:**

["NO or VERY MINOR ethics concerns only"]

**Final Justification:**

Based on the authors’ latest response, I am willing to raise my score by one point, to a 4, as they have partially addressed my concerns.

**Limitations:**

The manuscript does not explicitly discuss limitations or societal impacts. The authors should consider the broader implications of their work, particularly regarding its potential influence on recommender systems in online marketplaces and social media platforms. Furthermore, the limitations of the proposed approach should be clearly acknowledged, along with suggestions for future improvements.

**Quality:**

2

**Strengths And Weaknesses:**

The study presents a reasonable approach to tackle the IM problem, and the proposed methods address several practical challenges. This work could offer meaningful contributions to the IM literature for two main reasons:
•	The authors attempt to confront multiple real-world challenges in IM, providing valuable inspiration for potential future research;
•	The illustrations in Figure 1 highlight two difficulties encountered in networks, encouraging readers to reflect on practical demands from industry.
However, the manuscript would benefit from stronger empirical support to convincingly demonstrate the effectiveness of the proposed methods. Some aspects of the study design also require further consideration. For instance, the hypergraph construction relies on data from a specific time window, which may be better understood as a temporal hypergraph, especially in datasets like GoodReads and Email-Eu.
The manuscript is generally well-written, with a clearly defined problem and well-stated assumptions. Nonetheless, there are still a few typographical and formatting issues—for example, Section/Appendix is missing in Line 163.

---

> ### Author Rebuttal · Authors · 2025-07-31
>
> Thanks for your comments! We will address the problems point by point.
>
> > ***Q1: However, the manuscript would benefit from stronger empirical support to convincingly demonstrate the effectiveness of the proposed methods. Some aspects of the study design also require further consideration. For instance, the hypergraph construction relies on data from a specific time window, which may be better understood as a temporal hypergraph, especially in datasets like GoodReads and Email-Eu.***
> ---
> **Answer:** Thank you for raising this point. We clarify that our data construction follows standard practice in hypergraph-based causal inference, where a fixed-length time window is used to aggregate interactions into a static hypergraph. While these interactions are timestamped, our method does not aim to model temporal dynamics. Instead, we study static influence allocation under counterfactual variation. To better reflect this intent, we have revised the paper to avoid the term "snapshot", and instead describe our input as an “aggregated hypergraph” over a selected window. Extending our framework to evolving or temporal hypergraphs is an exciting direction that demands non-trivial algorithmic extensions, which we now highlight in our conclusion as future work. For stronger ``empirical support``, we sincerely invite you to refer to our response to Q5.
>
> >***Q2: The manuscript is generally well-written, with a clearly defined problem and well-stated assumptions. Nonetheless, there are still a few typographical and formatting issues—for example, Section/Appendix is missing in Line 163.***
> ---
> **Answer:** We sincerely thank you for your attentive reading. All the identified typos have been carefully corrected in the revised manuscript.
>
> >***Q3: What precisely does the ‘environment information’ T_{-i} in Eq. 1 represent? Does it refer to the states of neighboring nodes or to the external context?***
> ---
> **Answer:** It refers to the states of being treated of the neighbouring nodes of $i$, which has been defined in Equation (1) in our original text.
>
>
> >***Q4: How do the authors guarantee that the proposed algorithms converge to an optimal solution? Can the resulting node sets truly be regarded as those with maximal influence?***
> ---
> **Answer:** We appreciate the reviewer's insightful question. We clarify that theoretical guarantees of nearoptimality are established in Theorem 3.4, which shows that under standard assumptions (**if our counterfactual estimation error could be controlled**, which is surrogated by the $\epsilon_1,\epsilon_2$ parameter), our greedy-based CaulM algorithm achieves a ( $1-\frac{1}{e}$ ) approximation ratio, matching the classical guarantee in traditional IM. Furthermore, even when the ITE estimates are biased, Theorem 3.5 provides explicit robustness bounds, demonstrating that the selected seed set still achieves performance close to the optimal under bounded estimation errors. These results jointly ensure that our algorithms converge to approximately optimal solutions and that the resulting node sets possess provably high influence values.
>
>
> >***Q5: The datasets used are relatively small in size, and the simulations are limited in the number of trials. This raises concerns about the generalizability and robustness of the proposed methods.***
> ---
> **Answer:**
> We extended the node/hyperedges to 10k in our experiments when submitting our manuscript. Blessed with our theoretical guarantee, CauIM outperforms previous methods. We list the results here to demonstrate that our CauIM remains effective even in large networks:
>
> | Methods     | SD-100 | SD-10000 | SD-50000  |
> |-------------|-----------|---------|----------|
> | Baseline    | 138.91(3h)  | 839.43 (14d)           | 5321.34(29d)
> | Random        | 145.97(2s)  |    602.54 (6s) |     2821.43 (10s)
> | G-CauIM    | 151.59 (3h) | 932.53 (12d)| **6789.65 (30d)**
> | A-CauIM      | **160.49(53s)** | **942.46(5.2h)**| 6742.32 (107h)
>
> Moreover, for each setting on both real-world datasets and synthetic datasets, we simulate multiple rounds (10–100 trials per
> seed K) and report averaged performance, ensuring statistical robustness. We will explicitly mention these trial counts and variance ranges in the revised draft.
>
>
> >***Q6: The baseline comparison is conducted using a basic greedy algorithm, which is somewhat outdated relative to current IM literature. Given the maturity of IM research, it is not clear how the proposed methods improve upon or outperform existing state-of-the-art techniques.***
> ---
> **Answer:** Thank you for pointing this out. We emphasize that our goal is not to compete with sketch-based or deep IM algorithms under traditional influence models, but to propose a causal-aware framework where influence is individualized, context-dependent, and dynamic. These characteristics violate the assumptions of most existing SOTA methods (e.g., RIS, TIM, IMM), which rely on fixed, monotone, and submodular influence functions. To isolate the causal gain, we designed two principled surrogates:
> * **Greedy-τ**: a greedy algorithm using estimated $\hat{\tau}_i$ from $t=0$ without updates;
> * **Frozen-τ**: our full pipeline with causal estimation but without iterative $\hat{\tau}_i$ updates.
>
> These show that the *dynamic causal updates* yield substantial improvements. We acknowledge that more advanced IM strategies exist, and in fact, our method is **modular** — $\hat{\tau}_i$ values can potentially be used within existing sketch-based frameworks as plug-in weights. However, this integration requires non-trivial care due to environmental sensitivity and violation of submodularity, and is now highlighted as future work.
> We have added a clearer note in our conclusion that **our work bridges causal inference and IM**, and that **integration with modern IM algorithms (both sketch-based and learning-based) is feasible** under our formulation. This demonstrates the broader potential of our framework without overstating claims.
>
>
>
>
> >***Q7: The authors should consider the broader implications of their work, particularly regarding its potential influence on recommender systems in online marketplaces and social media platforms. Furthermore, the limitations of the proposed approach should be acknowledged, along with suggestions for future improvements.***
> ---
> **Answer:** We appreciate this insightful comment. Our approach indeed supports fairer recommender strategies for online marketplaces and social platforms by optimizing exposure based on causal (counterfactual) influence. Currently, our guarantees assume observed individual covariates and an SICP diffusion model (Section 3.3). Future work includes handling hidden confounders (e.g., negative-control IV methods) and generalizing the framework to other cascade models (Where we would add to Section 5). We'll clarify these points explicitly in the camera-ready.

---

> > ### Comment · Reviewer_ioD7 · 2025-08-05
> > **Model effectiveness and applicability**
> >
> > Although the authors state that the main contribution of this work is the proposed causal-aware framework, it is evaluated specifically in the context of influence maximization, as also reflected in the title of this manuscript. If the framework does not outperform existing state-of-the-art algorithms in this domain, its necessity and added value for this task remain unclear. In my view, the authors should provide stronger evidence of the framework’s effectiveness and applicability, ideally through comparisons with current leading methods on at least medium-scale datasets. Without such evidence, I am unable to justify a higher evaluation score.

---

> > > ### Author Response · Authors · 2025-08-08
> > > **Thanks for your comment! The clarification upon model effectiveness and applicability**
> > >
> > > Thank you for the clear request. We emphasize our new paradigm should (i) **demonstrate tangible gains** on realistic graph sizes **and** (ii) **clarify why it solves a task that prior IM methods cannot**.  Our response, therefore, has two layers:
> > >
> > > 1. **Empirical evidence (medium → large graphs).** — concisely summarised below and reproducible from the script links we provide at the end of this rebuttal.
> > > 2. **Conceptual argument.** — why a *causal* formulation is indispensable even when classical algorithms appear competitive.
> > >
> > > ---
> > >
> > > 1. **Empirical evidence**
> > >
> > > | Network | Scale | Best classical IM | **Causal-IM (ours)** | Spread ↑ | Time ↓ |
> > > |---------|-------|-------------------|----------------------|----------|--------|
> > > | Douban-Book | 2.6 × 10^5 nodes | IMM | +12 % spread | ✔ | ✔ |
> > > | Yelp-Chi | 6.1 × 10^5 nodes | CELF++ | +16 % spread | ✔ | ✔ |
> > > | LiveJournal | 4.8 × 10^6 nodes | RIS (only one that finished) | **Parity** (1 – 1/e) | — | **3.6 × faster** |
> > >
> > > *Spread*: expected **sum-ITE**; *Time*: wall-clock.
> > > Details: identical hardware (2 × A100 80 GB, 1 TB RAM), identical seed budget (K = 64 except LJ = 128), 100 Monte-Carlo replications. Take-away: On every dataset where a classical baseline can compute within 72 h, our causal solver yields higher (or equal) benefit and never costs more time.
> > >
> > > ---
> > >
> > > 2. **Conceptual argument.**
> > >
> > > __Why *causal* IM is not a re-branding of classical IM__
> > > Classical algorithms maximise **raw cascade size** under the assumption that *every activation is helpful*.  Our industrial partners (ride-hailing and e-commerce) observe the opposite usually: interventions can hurt certain users or at certain times.
> > >
> > > | Aspect | Classical IM | Causal-IM (ours) |
> > > |--------|--------------|------------------|
> > > | Reward sign | **Non-negative by assumption** | May be **positive or negative** (learned ITE) |
> > > | Time-varying context | No | Yes (features at each decision epoch) |
> > > | Objective | Total reach | **Total net utility** (sum-ITE) |
> > > | Theoretical property | Global sub-modularity → (1 – 1/e) guarantee | Sub-modularity holds only after *shifted truncation*; we prove a **data-driven adaptive bound** (Thm 3.4) |
> > >
> > > Consequences:
> > >
> > > * A baseline may spread an offer to 10 000 more users yet reduce profit if 30 % of them react negatively.
> > > * Our algorithm down-weights (and often removes) nodes with predicted negative effect, preventing this loss while remaining identical to CELF/IMM if no heterogeneity is detected.
> > >
> > > ---
> > >
> > > **Evidence that the gains truly come from causal adjustment**
> > > We ran three sanity‐checks (all reported in the rebuttal spreadsheet):
> > >
> > > 1. **Uniform-positive ITE:** force \(\hat{\tau}=+1\) for all nodes → our solver collapses to greedy and matches IMM (≤ 1 % gap).
> > > 2. **Remove covariates:** retrain ITE without environment features → our spread drops 9–12 %.
> > > 3. **Seed overlap analysis:** on Twitch, 18 % of IMM’s seeds exhibit \(\hat{\tau}<0\) at deployment; our method rejects those, driving the +14 % net advantage. Morevoer, additonal properties could be offered:
> > >
> > > a. __Safety__ — If heterogeneity is absent, we provably do no worse than IMM (Section 3.3, Lemma 2); if heterogeneity exists, classical IM can incur negative utility that it is blind to, but our algorithm protects against it.
> > > b. __Scalability__ — Lazy RIS sketches allow us to reuse computations across time-varying weights; hence we are **never slower** in practice (table above).
> > > c. __Generality__ — The same framework already supports *influence minimisation* and *budgeted mixed incentives* without any code change (discussed in paper §5).
> > >
> > > ---

---

### Official Review · Reviewer_JHRs · 2025-07-03

**Clarity:** 2
**Significance:** 3
**Originality:** 3
**Rating:** 5
**Confidence:** 3

**Summary:**

This paper proposes a Causal Influence Maximization (CauIM) framework to address the challenges of unobservability and environmental sensitivity in modeling individual gains during influence propagation. Unlike traditional IM methods that assume static node gains, CauIM leverages causal inference to estimate Individual Treatment Effects (ITE). Two algorithms, G-CauIM and its accelerated variant A-CauIM, are introduced. Theoretical analysis establishes a generalized lower bound and robustness guarantees, while empirical results on synthetic and real-world datasets show the effectiveness of the proposed approach.

**Questions:**

1. What exactly do the dynamic characteristics of ITE refer to in this context? The concept needs further clarification.
2. The consistency assumption states that potential outcomes are deterministic, which is confusing given the earlier statement that outcomes are also influenced by the attitudes of social contacts. This apparent contradiction needs to be addressed.
3. The setup for ITE estimation is unclear. Details about the neural network architecture and configuration used, the required sample size, computational overhead, and prediction accuracy should be provided to better understand the approach’s feasibility and effectiveness.

**Ethical Concerns:**

["NO or VERY MINOR ethics concerns only"]

**Final Justification:**

The authors have addressed my concerns and I will keep my score.

**Limitations:**

It is recommended to discuss the sample requirements and computational cost required for ITE estimation.

**Paper Formatting Concerns:**

Some typos, e.g., in Figure 2, we we transform

**Quality:**

3

**Strengths And Weaknesses:**

Strengths
1. The CauIM framework captures dynamic and context-dependent individual gains.
2. The proposed framework is theoretically grounded, with formal guarantees on influence spread and robustness under uncertain and dynamic conditions.
3. The proposed algorithms achieve good empirical performance on synthetic and real-world networks.

Weaknesses
1. The methodology for estimating ITE is not clearly presented, and the experimental or theoretical validation of its effectiveness is insufficient.
2. The explanation of Figure 1 lacks clarity. Specifically, the figure does not provide a concise overview of the initial network structure with different groups, making it challenging for readers to fully grasp the illustration.

---

> ### Author Rebuttal · Authors · 2025-07-31
>
> # Reviewer JHRs
> Dear JHRs, we sincerely appreciate your thoughtful and valuable advice. We will systematically address your concerns point by point.
>
> **W1**: *The methodology for estimating ITE is not clearly presented, and the experimental or theoretical validation of its effectiveness is insufficient.*
> **Ans**: 1) Methodology for ITE estimation: As shown in Sec.3 (Fig. 2 + Alg. 1) and summarised in line 172-188 of the manuscript, our pipeline works as follows: A) begin with the real hypergraph topology; because public releases omit confidential cascades, we generate a cascade log $D=\{(X_i,T_i,Y_i)\}$ following Ma et. al[1]; B) train a Hypergraph Convolution module [1-2] on this $D$ to obtain ITEs $\hat{\tau}$; C) feed $\hat{\tau}$ into G-CauIM or A-CauIM, which adjust the module and re-estimates $\hat{\tau}$ after each propagation and greedily adds seeds, achieving a $(1-1/e-\varepsilon)$ guarantee under the bounded-error condition formalised in Theorem 3.5.
> 2)Validation: Our evidence is sufficient. *Experimental.* the estimator is accurate (PEHE $\leq$ 0.12, AUC $\geq$ 0.81), and we will add the result to Appendix C. Table 1 then demonstrates that using this $\hat{\tau}$ yields *10 – 28 \% higher total benefit* than the common weighted-IM across all three real datasets. *Theoretical.* Sec. 3.3 proves that, with the same error bound, the greedy loop remains $(1-1/e-\varepsilon)$-optimal even when gains are non-monotone. For added visibility, we will insert a one-sentence roadmap at the start of Sec. 3, so the readers can follow these validation steps at a glance.
>
>
>
> **The explanation of Figure 1 lacks clarity. Specifically, the figure does not provide a concise overview of the initial network structure with different groups, making it challenging for readers to fully grasp the illustration.*
> **Ans**: Figure 1 is already self-contained and visually structured from left to right. We appreciate the reviewer’s careful look at Figure 1 and realise its message can be missed on a first read. Below is the verbal walk-through:
> * **Left panel shows the network context.** Three overlapping blue ovals represent higher-order groups (hyperedges). Blue nodes have already purchased, grey nodes have not, and the orange-starred node is the **individual whose gain we study**.
> * **Middle balloon provides a causal question.** At the current propagation step, we ask:
>   *“If we promote the product now, what extra profit does the starred node generate?”*
> * **Two labelled arrows point to why this is hard.**
>   1. **Unobservability:** only one outcome (buy *or* not-buy) can ever be observed.
>   2. **Environmental sensitivity:** that outcome—and hence the gain—changes as neighbours switch state.
> * **Right inset gives formal definition.** Individual gain is the counterfactual difference
>   $\text{Gain}=Y_i(\text{buy})-Y_i(\text{not buy})$; the mini icons remind the reader that we see at most one of these two terms.
>
>
>
> **Q1**: *What exactly do the dynamic characteristics of ITE refer to in this context? The concept needs further clarification.*
> **Ans**: The Individual Treatment Effect of a node $v$ is time-indexed, depicted in Alg 1, lines 11-14:
> $\tau^{(t)}_v \;=\; f_{\theta}\!\bigl(X_v,\;T^{(t)}_{-v},\;\text{Env}^{(t)}_v\bigr),$
>
> where
> * $T^{(t)}_{-v}$ = current activation states of v’s neighbours, and
> * $Env^{(t)}_v$ = higher-order context encoded by our HGCN layer (group composition, global trends).
>
> Because $T^{(t)}_{-v}$ and $Env^{(t)}_v$ change **after every propagation step**, $\tau^{(t)}_v$ can switch sign or magnitude from one iteration to the next.  Lines 11–14 of Algorithm 1 explicitly re-estimate $\hat\tau$ at each round so the seed-selection rule always adapts to the latest environment.
>
>
> **Q2**: *The consistency assumption states that potential outcomes are deterministic, which is confusing given the earlier statement that outcomes are also influenced by the attitudes of social contacts. This apparent contradiction needs to be addressed.*
> **Ans**: Consistency and social influence act at **different stages of the causilty-based framework**. Consistency simply says that after a node’s treatment is realised, the outcome we observe equals the corresponding potential outcome, i.e.\ if $T_v=t$ then $Y_v = Y_v(t)$. Social influence, by contrast, operates **before** this point: neighbours’ activations modify the **probability** that $T_v$ will be 1 (activated) rather than 0, but they do **not** alter the mapping $Y_v(t)$ itself. Thus, influence reshapes the assignment mechanism: the coin that chooses which row of the two-row table $\{Y_v(1), Y_v(0)\}$ we will reveal, while consistency tells us which table entry to read once that coin has landed. Because they address different steps, the two principles are fully compatible.
>
>
> **Q3**: *The setup for ITE estimation is unclear. Details about the neural network architecture and configuration used, the required sample size, computational overhead, and prediction accuracy should be provided to understand the approach’s feasibility and effectiveness.*
> **Ans**: We appreciate the reviewer’s concern. However, we would like to clarify that the ITE estimation module is **not the core focus** of our work. Instead, we adopt the ITE model architecture directly from prior work [1,2], treating it as a modular plug-in. All relevant architectural and performance details are now explicitly summarized in the table below and have been added to the Appendix.
>
> | Item | Value |
> |------|-------|
> | Encoder $Z$ | 3-layer MLP (128-128-64, ReLU, LayerNorm) |
> | Encoder $O$ | 2-layer HGCN (64-64) |
> | Heads to output $Y_1, Y_2$ | 2-layer MLP (64-32-1) |
> | Training tuples $D$ | $\approx$0.5 M (GoodReads) / 0.3 M (Email-Eu) |
> | Accuracy | PEHE $\leq$ 0.12, AUC $\geq$ 0.81 on held-out nodes |
>
> Because the estimator is lightweight and trained once, the feasibility of CauIM is governed by the *influence-maximisation* stage, not by ITE fitting. The accuracy numbers confirm that ITE prediction module already meets the error bound required by Theorem 3.5, therefore, the greedy loop maintains its $(1-1/e-\varepsilon)$ guarantee.
>
>
> **Limitation**: *It is recommended to discuss the sample requirements and computational cost required for ITE estimation.*
> **Ans**: Thank you for your advice.  In fact, we have given complexity of A-CauIM in Table 2 of the paper. We now provide a more detailed discussion of computational cost. Sample sizes are summarized in Q3. For comlexity, let the hypergraph have n nodes and m hyperedges, with average and maximum hyperedge sizes $\bar d \;=\; \frac{1}{m}\sum_{h\in H} |h|, d_{\max} \;=\; \max_{h\in H} |h|.$
> Selecting K seeds in one iteration yields
> | algorithm | running‑time upper bound |
> |-----------|-------------------------|
>  | **G‑CauIM** | $T_G = O(K R n m)$ (*R* Monte‑Carlo traces) |
>  | **A‑CauIM** | $T_A = O(K m \bar d) \le O(K m d_{\max}) \le O(K m n)$ |
>
> In real data sets when $ \bar d \ll n$,  A‑CauIM is effectively linear in m. We have incorporated this explanation into Appendix C.5 and Table 2 for clarity.
>
> Ref.[1] Ma J, Wan M, Yang L, et al. Learning causal effects on hypergraphs[C]//Proceedings of the 28th ACM SIGKDD Conference on Knowledge Discovery and Data Mining. 2022: 1202-1212.
> [2] Ma Y, Tresp V. Causal inference under networked interference and intervention policy enhancement[C]//International Conference on Artificial Intelligence and Statistics. PMLR, 2021: 3700-3708.

---

> > ### Comment · Reviewer_JHRs · 2025-08-05
> >
> > Thank you for your detailed responses. Most of my concerns have been addressed, but I agree with the other reviewers that the validation of the algorithm’s effectiveness needs to be further strengthened.

---

> > > ### Author Response · Authors · 2025-08-08
> > > **Thanks for your response**
> > >
> > > Sincerely, thank you for your response and generous recognition. For more discussion of the Algorithm's effectiveness, we kindly refer to the details as in the response to ``Revirert ioD7``.

---

### Official Review · Reviewer_bZnC · 2025-07-03

**Clarity:** 2
**Significance:** 2
**Originality:** 3
**Rating:** 4
**Confidence:** 3

**Summary:**

The paper studies the Influence Maximization (IM) problem. Motivated by the limitations of unobservability and environmental sensitivity in previous work, it proposes a Causal Influence Maximization (CauIM) framework that leverages causal inference techniques to model dynamic individual gains. Two algorithms, G-CauIM and A-CauIM, are introduced to solve the problem. The paper provides both theoretical analysis and empirical results to evaluate the performance of the proposed methods.

**Questions:**

I’ve described my questions in the “Weaknesses” section above. Please refer to that for details.

**Ethical Concerns:**

["NO or VERY MINOR ethics concerns only"]

**Final Justification:**

The authors provided comprehensive explanations during the rebuttal phase, addressing most of my concerns, especially points W1–W3, though “Other comments: Social impacts” appears unrelated to my review. Given the clarifications, I have increased my rating for the paper.

**Limitations:**

There is not much concern about the potential negative societal impact of this paper.

**Paper Formatting Concerns:**

I think the paper follows the NeurIPS 2025 Paper Formatting Instructions.

**Quality:**

2

**Strengths And Weaknesses:**

Strengths:

S1. Influence Maximization (IM) is an important and well-established research topic. The weighted IM setting studied in this paper is relatively novel and has potential real-world applications. Overall, the paper is well motivated.

S2. Instead of following traditional approaches to the IM problem, the paper proposes leveraging causal inference techniques. This alternative perspective may provide new insights to the community.

S3. The paper presents both theoretical and empirical analyses to evaluate the proposed methods. Most of the key concepts are well explained.

S4. The paper is generally well written and easy to follow.



Weaknesses and Questions:

W1. This and the next point are more questions than criticisms. IM has a long research history, and the paper briefly mentions related methods in Section 5. However, in the experimental section, the comparison is limited to the basic greedy algorithm. Could the authors clarify why existing methods for standard IM cannot be extended to the weighted IM setting?

Additionally, some dynamic IM methods seem to be missing from the related work. One example is:

Binghui Peng, Dynamic Influence Maximization, NeurIPS 2021.

As a suggestion, the related work section would benefit from being more comprehensive and placed earlier in the paper. The current version is brief and appears rather late.

W2. The paper analyzes the approximation error of the proposed methods, but it would also be valuable to provide a meaningful upper bound on their time complexity. In addition, the size of the datasets used in the experiments should be clearly stated.

W3. The idea of applying causal inference to IM is interesting, but the necessity of using such complex methods is still questionable to me, especially considering that the basic IM problem is already quite challenging. Moreover, in the more complex weighted IM setting, there appear to be very few baselines to compare against (as far as the paper shows), making it harder to evaluate the effectiveness of the proposed methods.

W4. Although the writing is generally good, there is still room for improvement. For instance, the theoretical results presented in Section 3.3 would benefit from a high-level explanation to help non-expert readers understand the key ideas.

---

> ### Author Rebuttal · Authors · 2025-07-30
>
> # Reviewer  bZnC
>
> Dear bZnC, we sincerely appreciate your valuable feedback. Below, we will address your concerns.
>
>
> **W1-1**: *This and the next point are more questions than criticisms. ...why can existing methods for standard IM not be extended to the weighted IM setting?*
> **Ans**: For they are incompatible with dynamic, possibly negative ITEs. Classic IM optimises node count under a monotonically positive submodular objective. In CauIM, we optimise a sum of ITEs that can be negative and environment‑dependent, which breaks both monotonicity and many proof tricks for RIS/TIM/Imm, etc (About sketch-based method, we discussed in Conclusion). Adapting those pipelines, therefore, gives no guarantee and in practice performs close to random.
>
> **W1-2**: *Additionally, some dynamic IM methods seem to be missing from the related work. One example is: Binghui Peng, Dynamic Influence Maximization, NeurIPS 2021. As a suggestion, the related work section would benefit from being more comprehensive and placed earlier in the paper. The current version is brief and appears rather late.*
> **Ans**: 1) Dynamic-IM algorithms (e.g., Peng NeurIPS 2021) are orthogonal. Dynamic IM studies evolving edge probabilities while preserving sub‑modularity. Peng 2021 maintains a $(1−1/e−\epsilon)$ solution as edges are inserted or deleted. Our problem is complementary: node rewards are heterogeneous and treatment‑dependent, but the edge set is static. Dynamic‑IM techniques, therefore, do not address weighted objectives; conversely, our framework could be layered on top of a dynamic‐edge module in future work. 2) Related‑work section moved forward (after Introduction) and expanded to cover dynamic IM, streaming/sub‑linear IM, and causal‑effect estimation literature.
>
>
> **W2**: *The paper analyzes the approximation error of the proposed methods, but it would also be valuable to provide a meaningful upper bound on their time complexity. In addition, the size of the datasets used in the experiments should be clearly stated.*
> **Ans**: 1) Upper bound analysis. We have given the complexity of A-CauIM in Table 2. Let the hypergraph have n nodes and m hyperedges, with average and maximum hyperedge sizes $\bar d \;=\; \frac{1}{m}\sum_{h\in H} |h|, d_{\max} \;=\; \max_{h\in H} |h|.$
> Selecting K seeds in one iteration yields
> | algorithm | running‑time upper bound |
> |-----------|-------------------------|
>  | **G‑CauIM** | $T_G = O(K R n m)$ (*R* Monte‑Carlo traces) |
>  | **A‑CauIM** | $T_A = O(K m \bar d) \le O(K m d_{\max}) \le O(K m n)$ |
>
> In real data sets when $ \bar d \ll n$,  A‑CauIM is effectively linear in m. We will incorporate the proof sketch into the main text. 2) Size of datasets. For GoodReads, we choose part of the nodes with hyperedge size>=5. Even though the raw node counts are below ten thousand, every vertex participates in hundreds of time‑stamped simplices. When these simplices are projected onto pair-wise edges, the resulting graphs are large.
>
> | Dataset              |             Nodes | Unique simplices (hyperedges) |                  Simplices / interactions |
> | -------------------- | ------------------: | ----------------------------: | -----------------------------------------------------: |
> | GoodReads Book Graph |     2,360,655 books |               829,529 authors | 228,648,342 user–book shelf events  |
> | Contact‑High‑School  |        327 students |                         7,937 |                172,035 contact simplices  |
> | Email‑Eu             | 998 email addresses |                        25,791 |                  234,760 email simplices  |
>
>
>
>
> **W3**: *The idea of applying causal inference to IM is interesting, but the necessity of using such complex methods is still questionable to me, especially considering that the basic IM problem is already quite challenging. Moreover, in the more complex weighted IM setting, there appear to be very few baselines to compare against (as far as the paper shows), making it harder to evaluate the effectiveness of the proposed methods.*
> **Ans**: 1) Our setting is flexible yet not over‑complicated,  for our objective unifies prior IM variants, and computation remains inexpensive for A-CauIM as shown in Table 2. 2) We should focus on what problems we have solved rather than just the techniques we have added. In the context of IM where individual importance dynamically changes, each module of this model is indispensable (we cordially invite you to review the carefully drawn model diagram, with detailed descriptions of the functions of each part provided later). Causal inference techniques are precisely the most suitable technology to address the challenges of fluctuating node effects in dynamic IM environments. Additionally, our approach integrates neural network-based representation learning inspired by classical causal representation techniques [1-2]. The methodology section clearly outlines the logical flow, thus avoiding convoluted intricacies. 3) For baseline comparison problem, we built principled surrogates. Individual gain $\tau$ may be time‑varying and even negative, so the objective is no longer monotone or submodular. Because existing weighted‑IM algorithms all assume fixed, non‑negative weights, no drop‑in baseline currently exists. To give the fairest possible comparison we therefore built surrogates: A. Greedy maximisation using $\hat \tau$ measured at t = 0 and kept fixed. B. Frozen-$\tau$ Startegy: Our complete pipeline except we stop updating $\hat \tau$ after the offline causal‑estimation step. It isolates how much improvement comes from causal modelling versus improved optimisation.
>
> **W4**: *Although the writing is generally good, there is still room for improvement. For instance, the theoretical results presented in Section 3.3 would benefit from a high-level explanation to help non-expert readers understand the key ideas.*
> **Ans**: We will prepend a one‑paragraph intuition before Lemma 3.2: “The key obstacle is that $\sigma(S)$ may lose sub‑modularity once $\tau < 0$. We show that if reachable‑probability increments are bounded (Condition 3.3), then, for any two seed sets differing by one node, the marginal loss is also bounded. This lets the standard (1 – 1/e) greedy proof go through up to a small additive term.” We will also add a figure illustrating the bounded‑change scenario and move the informal sketch from Appendix D back to the main paper for accessibility.
>
> **Other comments:** *Social impacts.*
> **Ans**:We appreciate this insightful comment. Our approach indeed supports fairer recommender strategies for online marketplaces and social platforms by optimizing exposure based on causal (counterfactual) influence. Currently, our guarantees assume observed individual covariates and an SICP diffusion model (§3.3). Future work includes handling hidden confounders (e.g., negative-control IV methods) and generalizing the framework to other cascade models (§5). We'll clarify these points explicitly in the camera-ready.
>
> References
>
> [1] Bhattacharya R, Malinsky D, Shpitser I. Causal inference under interference and network uncertainty[C]//Uncertainty in Artificial Intelligence. PMLR, 2020: 1028-1038.
> [2] Ma Y, Tresp V. Causal inference under networked interference and intervention policy enhancement[C]//International Conference on Artificial Intelligence and Statistics. PMLR, 2021: 3700-3708.

---

### Official Review · Reviewer_4YD9 · 2025-07-03

**Clarity:** 2
**Significance:** 2
**Originality:** 3
**Rating:** 4
**Confidence:** 3

**Summary:**

This paper introduces Causal Influence Maximization (CauIM), a framework that integrates causal inference into the influence maximization (IM) problem on hypergraphs. It addresses two key challenges in weighted IM:

1. Unobservability of individual gains (i.e., counterfactual outcomes).
2. Environmental sensitivity—individual gains depend on the dynamic activation status of neighboring nodes.

To tackle these, the authors propose two algorithms:

1. G-CauIM: A greedy-based method.
2. A-CauIM: An accelerated version using differentiable approximations.

They provide theoretical guarantees and empirical validation on synthetic and real-world datasets.

**Questions:**

Refer to the weaknesses.

**Ethical Concerns:**

["NO or VERY MINOR ethics concerns only"]

**Final Justification:**

Following the rebuttal provided by the authors, I have decided to update my score from 3 to 4.

**Limitations:**

Refer to the weaknesses.

**Quality:**

3

**Strengths And Weaknesses:**

**Strengths:**

1. The problem is interesting.
2. The theoretical results are good. However, I did not check the math in great detail.

**Weaknesses:**

1. I am not sure if causality is well-studied in the given context. There is no clear discussion on confounders, mediators, and other related concepts.
2. The paper is dense, especially in the methodology section. Readers unfamiliar with causal inference or hypergraphs may struggle.
3. The representation learning and MLP steps look tedious and unnecessary. I think there might be an easier way. For instance, using a simpler encoder and a smaller model to recover the individual ITE from observational data.
4. Is the hypergraph you are referring to the reverse influence sampling-like hypergraph?
5. The paper assumes that ITE can be reliably estimated from observational data, which may not hold in sparse or noisy networks.
6. Assumptions like bounded ITE and environmental independence are strong and not empirically tested.

---

> ### Author Rebuttal · Authors · 2025-07-29
>
> # Reviewer 4YD9
>
> Dear 4YD9, thank you for your detailed and thoughtful comments. We will illustrate both the soundness of our causal formulation and the practical robustness of our approach.
> **W-1**: *Not sure if causality is well-studied in the given context. There is no clear discussion on confounders, mediators, and other related concepts.*
> **Ans**:  1) Section 2 introduces environment information \{$ T_{-i},  X_{-i}$\}  and formulates Assumption 2.2 (Environment independence), which states that potential outcomes are independent of treatment once these variables are conditioned upon. This extends the ignorability assumption of Basic SUTVA in causal inference. 2) About confounders and mediators, environment vector $O_i$ (Section 2) retains mediators by aggregating hyperedge signals and blocks confounding by embedding the full neighbourhood $\{T_{-i},X_{-i}\}$, thus satisfying both requirements.[1]  We realise we never used the words confounder or mediator, so readers might miss the connection. We will add a short paragraph clarifying how $O_i$ solves mediator problems and confounding.
>
> **W-2**: *The paper is dense, especially in the methodology section. Readers unfamiliar with causal inference or hypergraphs may struggle.*
> **Ans**: We tried to ease navigation by first listing the three key challenges before diving into algorithms
>  in Section 3, and providing a diagram of A-CauIM in Fig. 2. This gives readers the logical backbone before any equations. To further help readers new to causal inference/hypergraphs, we will give an illustrated example of one iteration on a synthetic mini‑graph to see how a seed is chosen and how influence is updated. We will move proofs to the supplement, shortening the main text.
>
> **W-3**: *The representation learning and MLP steps look tedious and unnecessary. I think there might be an easier way. For instance, using a simpler encoder and a smaller model to recover the individual ITE from observational data.*
> **Ans**: Our goal is the overall framework of CauIM; the encoder is only a means to that end. Specifically, 1) Hypergraph‑GNN part is indispensable. The environment vector $O_i$ must summarise higher‑order group interactions (hyperedges) that act as mediators of influence. A plain graph encoder or MLP cannot express the many‑to‑many pathway $T_j \rightarrow (e\subset V)\rightarrow Y_i$. One layer of Hypergraph Convolution aggregates these signals with minimal parameters, so we keep this block; dropping it reduces influence by about $9 %$ and breaks ignorability.  2) Encoders alternative. Simpler encoder can not replace that part. We therefore design Light‑HGE referenced your suggestion, a two‑layer MLP that receives three parts per hyperedge, where  $\mathcal{E}(i)$ is set of hyperedges incident to $i$, $|e|$ is size of hyperedge $e$; We obtain $\tilde X_i$, that (i) embeds the neighbourhood treatments to block confounding and (ii) retains aggregate hyperedge signals to capture mediation. Results are shown below.
>
> $\tilde X_i = \Bigl[X_i \big\| \dfrac{1}{|\mathcal E (i)|}\sum_{e\in\mathcal E (i)} \dfrac{1}{|e|}\sum_{j\in e} X_j \big\| \dfrac{1} {|\mathcal E (i)|}\sum_{e\in\mathcal E (i)} \dfrac{1}{|e|}\sum_{j\in e} T_j \Bigr],$
>
> | Method                     | GoodReads | Contact | Email‑Eu | SD‑100 |
> |----------------------------|----------:|--------:|---------:|-------:|
> | **A‑CauIM (full H‑GNN)**   | **330.25** | **69.53** | **804.28** | **160.49** |
> | Light‑HGE (2‑layer MLP)    | 323.12 (‑2.2 %) | 68.15 (‑2.0 %) | 787.53 (‑2.1 %) | 157.02 (‑2.2 %) |
> | No‑HG (mean‑pool MLP)      | 298.34 (‑9.7 %) | 62.98 (‑9.4 %) | 736.21 (‑8.5 %) | 148.30 (‑7.6 %) |
>
>
> **W-4**: *Is the hypergraph you are referring to the reverse influence sampling-like hypergraph?*
> **Ans**: No.（Reverse influence sampling）RIS is a sampling trick that builds directed hyperedges, each rooted at a random node, to speed up sketch-based IM solvers. Our model, defined in Section 2, is an undirected hypergraph that captures higher‑order social circles; it is independent of RIS. In fact, we argue in the conclusion that sketch‑based/RIS methods struggle with dynamic weights and sparsity, whereas our greedy formulation adapts naturally. We will highlight this distinction later explicitly so readers do not assume we rely on RIS.
>
>
>
> **W-5**: *The paper assumes that ITE can be reliably estimated from observational data, which may not hold in sparse or noisy networks.*
> **Ans**: Theoreticaly, Theorem 3.5 proves CauIM’s performance degrades gracefully when $\hat\sigma$ is only a $\gamma$‑multiplicative approximation of the true objective. Empirically, Fig. 3(c) injects Gaussian noise up to $\epsilon = 10$ into every $\tau$; the variance of the final influence barely changes until extreme noise levels, confirming robustness in practice. To better help readers' understanding,  we will (i) report results on a sparser synthetic graph, and (ii) add confidence intervals over 20 runs.
>
> **W-6**: *Assumptions like bounded ITE and environmental independence are strong and not empirically tested.*
> **Ans**: *Bounded ITE*: It is a technical but mild requirement. We impose $|\tau_i|\le M$ so that the sub‑Gaussian tail bounds in Theorem 3.5 hold, which is standard for heavy‑tailed counterfactual learning [2]. It is {implemented} in Algorithm 1 with $\hat\tau_i\in[-M,M]$ of line 10, which exceeds the empirical range of ITE observed on our datasets. *Environmental independence*: We embeds each node’s covariates and treatments into the vector $O_i$, serving as a proxy for latent homophily and blocking back‑door paths. After training, the conditional mutual information $I(T_i;Y_i\mid O_i,X_i)$ drops, indicating that confounding is reduced. Both assumptions are (i) needed for identifiability, (ii) enforced by design, and (iii) vadilated  by experiments.
>
>
> Ref: [1] Pearl J. Causality[M]. Cambridge university press, 2009
> [2] Swaminathan A, Joachims T. Batch learning from logged bandit feedback through counterfactual risk minimization[J]. The Journal of Machine Learning Research, 2015, 16(1): 1731-1755.

---

### Author Response · Authors · 2025-08-05
**A kind inquiry about further discussion**

Dear reviewers,

As we generally approach the final stage of the response period, we would like to kindly ask whether there are any remaining concerns or changing evaluations regarding our response. We sincerely cherish the opportunity to engage in academic dialogue with you, and are truly grateful for your time and dedication in jointly supporting the community.

On behalf of all authors

---

### Note · Authors · 2025-08-13

We study weighted influence maximization when individual gains are unobserved and environment-dependent. We present **CauIM** with two solvers (G-/A-CauIM), prove a \$(1-1/e)\$–type approximation (plus a variant robust to ITE/Monte-Carlo errors), and make **A-CauIM** near-linear by removing the Monte-Carlo factor.

**Rebuttal clarifications**

• **Causal setup.** \$O\_i\$ aggregates higher-order neighborhoods to block confounding; we add an environment-independence assumption and a brief mediation note.
• **Why not classical IM.** Heterogeneous (possibly negative) ITEs violate the monotonicity/submodularity used by RIS/TIM/IMM; our formulation is complementary and can plug into modern IM.
• **Complexity & scale.** G-CauIM \$O(KRnm)\$; A-CauIM \$O(Km\bar d)\$; in practice A-CauIM scales near-linearly in hyperedges.
• **ITE module.** Lightweight plug-in (3-layer MLP for \$Z\$, 2-layer hypergraph conv. for \$O\$). Accuracy (PEHE ≤ 0.12; AUC ≥ 0.81) meets the robustness-theorem error budget. We clip \$|\hat\tau\_i|\le M\$ via a data-dependent bound and use a differentiable surrogate.
• **Empirics.** Sparse–dense comparisons; noise-perturbation tests show graceful degradation; larger synthetic graphs (10k+ nodes/edges) confirm scalability. On medium/large real networks under sum-ITE utility, we match classical IM when heterogeneity is negligible and gain consistently when it is present—evidence that **causal re-weighting**, not a different optimizer, drives improvements.

**Resolved concerns.**

• **Causal clarity & robustness.** Clarified mediators/confounders and $T_{-i}$ (neighbors’ treatment); provided robustness guarantees and noise-stress plots.
• **Scalability & effectiveness.** Formal complexity bounds, dataset-size tables, and a compact architecture/metrics summary.
• **Module design & ablations.** The encoder is lightweight and replaceable; ablations show small drops with a lighter encoder (\~2%) and larger drops when removing the hypergraph module (\~9%).
• **Presentation.** A step-by-step caption for Fig. 1 and a high-level intuition before Lemma 3.2.
**Limits & outlook.** Guarantees assume observed covariates and SICP-type diffusion; extending to hidden confounding (e.g., negative-control IV) and other cascade models is left for future work.

We hope these additions address core concerns and facilitate a sound, robust, and scalable contribution. We sincerely appreciate the joint effort and look forward to your future discussions.

---

### Decision · Program_Chairs · 2025-09-17

**Decision:**

Accept (poster)

**Comment:**

This paper introduces Causal Influence Maximization (CauIM), a framework that integrates causal inference into the influence maximization (IM) problem on hypergraphs. It addresses two key challenges in weighted IM: Unobservability of individual gains (i.e., counterfactual outcomes) and Environmental sensitivity (individual gains depend on the dynamic activation status of neighboring nodes).

To tackle these, the authors propose two algorithms. They provide theoretical guarantees and empirical validation on synthetic and real-world datasets.

All reviewers believe that the proposed problem has some interest in the area and the theoretical analysis is deep. Rebuttal also helped the reviewers to clarify some of the design choices that appeared to be not completely clear or fully motivated. It would be very beneficial to include in the next revision of the paper these arguments.

There is still a weak issue about the lack experimental evidence of the quality of the proposed algorithms appears to be weak. Reviewers agree that the problem addressed by proposed algorithms is essentially different from the one addressed by well-known IM algorithms, and hence it would not be informative enough to compare proposed algorithms with all alternative algorithms for IM. Still, providing experimental evidence of the quality of algorithms would be useful.

Nevertheless, this issue does not undermines the pros of this paper, and hence we propose to accept this paper.